# An Online Riemannian PCA for Stochastic Canonical Correlation Analysis

**Zihang Meng**[1][*] **Rudrasis Chakraborty**[2][*] **Vikas Singh**[1]
[1]University of Wisconsin-Madison   [2]Butlr
zihangm@cs.wisc.edu, rudrasischa@gmail.com, vsingh@biostat.wisc.edu

## Abstract

We present an efficient stochastic algorithm (RSG+) for canonical correlation analysis (CCA) using a reparametrization of the projection matrices. We show how this reparametrization (into structured matrices), simple in hindsight, directly presents an opportunity to repurpose/adjust mature techniques for numerical optimization on Riemannian manifolds. Our developments nicely complement existing methods for this problem which either require $O(d^3)$ time complexity per iteration with $O(\frac{1}{\sqrt{t}})$ convergence rate (where $d$ is the dimensionality) or only extract the top 1 component with $O(\frac{1}{t})$ convergence rate. In contrast, our algorithm offers an improvement: it achieves $O(d^2k)$ runtime complexity per iteration for extracting the top $k$ canonical components with $O(\frac{1}{t})$ convergence rate. While our paper focuses more on the formulation and the algorithm, our experiments show that the empirical behavior on common datasets is quite promising. We also explore a potential application in training fair models with missing sensitive attributes.

## 1 Introduction

Canonical correlation analysis (CCA) is a classical method for evaluating correlations between two sets of variables. It is commonly used in unsupervised multi-view learning, where the multiple views of the data may correspond to image, text, audio and so on, Rupnik and Shawe-Taylor [2010], Chaudhuri et al. [2009], Luo et al. [2015], and has been applied to manifold-valued data also Kim et al. [2014]. Classical formulations have also been extended to leverage advances in representation learning, for example, Andrew et al. [2013] showed how the CCA can be interfaced with deep neural networks enabling modern use cases. Many results over the last few years have used CCA or its variants for problems including measuring representational similarity in deep neural networks Morcos et al. [2018] and speech recognition Couture et al. [2019].

The goal in CCA is to find linear combinations within two random variables $\mathbf{X}$ and $\mathbf{Y}$ which have maximum correlation with each other. Formally, the CCA problem is defined as follows. Let $X \in \mathbf{R}^{N \times d_x}$ and $Y \in \mathbf{R}^{N \times d_y}$ be $N$ samples respectively drawn from pair of random variables $\mathbf{X}$ ($d_x$-variate random variable) and $\mathbf{Y}$ ($d_y$-variate random variable), with unknown joint probability distribution. The goal is to find the projection matrices $U \in \mathbf{R}^{d_x \times k}$ and $V \in \mathbf{R}^{d_y \times k}$, with $k \leq \min\{d_x, d_y\}$, such that the correlation is maximized:

$$\max_{U,V} F = \text{trace}\left(U^T C_{XY} V\right) \quad \text{s.t.} \quad\quad U^T C_X U = I_k, V^T C_Y V = I_k \quad\quad (1)$$

Here, $C_X = \frac{1}{N} X^T X$ and $C_Y = \frac{1}{N} Y^T Y$ are the sample covariance matrices, and $C_{XY} = \frac{1}{N} X^T Y$ denotes the sample cross-covariance.

The objective in (1) is the expected cross-correlation in the projected space and the constraints specify that different canonical components should be decorrelated. Let us define the whitened covariance

---

[*]Equal contribution

$T := C_X^{-1/2} C_{XY} C_Y^{-1/2}$ and $\Phi_k$ (and $\Psi_k$) contains the top-$k$ left (and right) singular vectors of $T$. It is known Golub and Zha [1992] that the optimum of (1) is achieved at $U^* = C_X^{-1/2}\Phi_k$, $V^* = C_Y^{-1/2}\Psi_k$. We can compute $U^*, V^*$ by applying a $k$-truncated SVD to $T$.

**Runtime and memory considerations.** The above procedure is simple but is only feasible when the data matrices are small. In modern applications, not only are the datasets large but also the dimension $d$ (let $d = \max\{d_x, d_y\}$) of each sample can be large, especially if representations are learned using deep models. As a result, the resource needs of the algorithm can be high. This has motivated the study of stochastic optimization routines for solving CCA, and many efficient strategies have been proposed. For example, Ge et al. [2016], Wang et al. [2016] present Empirical Risk Minimization (ERM) models which optimize the empirical objective. More recently, Gao et al. [2019], Bhatia et al. [2018], Arora et al. [2017] describe proposals that optimize the population objective. To summarize the approaches, if we are satisfied with the top 1 component of CCA, effective schemes with $O(\frac{1}{t})$ convergence rate are available by utilizing either extensions of the Oja's rule Oja [1982] to the generalized eigenvalue problem Bhatia et al. [2018] or the alternating SVRG algorithm Gao et al. [2019]. Otherwise, a stochastic approach will use an explicit whitening operation which can cost $d^3$ operations for each iteration Arora et al. [2017] and the convergence rate for the stochastic scheme depends on its specific steps and calculations, e.g., $O(\frac{1}{\sqrt{t}})$ in Arora et al. [2017] (Thm 2.3, pp 5).

**Observation.** Most approaches either directly optimize (1) or instead a reparameterized or regularized form Ge et al. [2016], Allen-Zhu and Li [2016], Arora et al. [2017]. Often, the search space for $U$ and $V$ corresponds to the entire $\mathbf{R}^{d \times k}$ (ignoring the constraints for the moment). But if the formulation could be cast in a form which involved approximately writing $U$ and $V$ as a product of structured matrices, we may be able to obtain specialized routines which are tailored to exploit those properties. Such a reformulation is not difficult to derive – where the matrices used to express $U$ and $V$ can be identified as objects that live in well studied geometric spaces. Then, utilizing the geometry of the space and borrowing relevant tools from differential geometry could lead to an efficient approximate scheme for top-$k$ CCA which optimizes the population objective in a streaming fashion.

**Contributions.** (a) First, we re-parameterize the top-$k$ CCA problem as an optimization problem on specific matrix manifolds, and show that it is equivalent to the original formulation in (1). (b) Informed by the geometry of the manifold, we derive stochastic gradient descent (SGD) algorithms for solving the re-parameterized problem with $O(d^2 k)$ cost per iteration and provide convergence rate guarantees. (c) This analysis gives a direct mechanism to obtain an upper bound on the number of iterations needed to guarantee an $\epsilon$ error w.r.t. the population objective for the CCA problem. (d) The algorithm works in a streaming manner so it easily scales to large datasets and we do not need to assume access to the full dataset at the outset. (e) We present empirical evidence for the standard CCA model and the DeepCCA setting Andrew et al. [2013], describing advantages and limitations.

## 2 Stochastic CCA: Reformulation, Algorithm and Analysis

Let us review the objective for CCA as given in (1). We denote $X \in \mathbf{R}^{N \times d_x}$ as the matrix consisting of the samples $\{\mathbf{x}_i\}$ drawn from a zero mean random variable $\mathbf{X} \sim \mathcal{X}$ and $Y \in \mathbf{R}^{N \times d_y}$ denotes the matrix consisting of samples $\{\mathbf{y}_i\}$ drawn from a zero mean random variable $\mathbf{Y} \sim \mathcal{Y}$. For simplicity, we assume that $d_x = d_y = d$ although the results hold for general $d_x$ and $d_y$. Also recall that $C_X$ (and $C_Y$ resp.) is the covariance matrix of $\mathbf{X}$ (and $\mathbf{Y}$ resp.) and $C_{XY}$ is the cross-covariance matrix between $\mathbf{X}$ and $\mathbf{Y}$. Let $U \in \mathbf{R}^{d \times k}$ ($V \in \mathbf{R}^{d \times k}$) be the matrix consisting of $\{\mathbf{u}_j\}$ ($\{\mathbf{v}_j\}$), where $(\{\mathbf{u}_j\}, \{\mathbf{v}_j\})$ are the canonical directions. The constraints in (1) are called *whitening constraints*.

**Reformulation:** In the CCA formulation, the matrices consisting of canonical correlation directions, i.e., $U$ and $V$, are unconstrained, hence the search space is the entire $\mathbf{R}^{d \times k}$. Now, we reformulate the CCA objective by reparameterizing $U$ and $V$. In order to do that, let us take a brief detour and recall the objective function of principal component analysis (PCA):

$$\widehat{U} = \underset{U'}{\arg\max} \quad \text{trace}(\widehat{R}) \qquad \text{subject to} \qquad \widehat{R} = U'^T C_X U'; \quad U'^T U' = I_k \qquad (2)$$

*Observe that by performing PCA and assigning $U = \widehat{U}\widehat{R}^{-1/2}$ in (1) (analogous for $V$ using $C_Y$), we can satisfy the whitening constraint.* Of course, writing $U = \widehat{U}\widehat{R}^{-1/2}$ does satisfy the whitening constraint, but such a $U$ (and $V$) will not maximize trace $\left(U^T C_{XY} V\right)$, the objective of (1). Hence,

additional work beyond the PCA solution is needed. Let us start from $\widehat{R}$ but relax the PCA solution by using an arbitrary $\widetilde{R}$ instead of diagonal $\widehat{R}$ (this will still satisfy the whitening constraint).

Write $U = \widetilde{U}\widetilde{R}$ with $\widetilde{U}^T\widetilde{U} = I_k$ and $\widetilde{R} \in \mathbf{R}^{k \times k}$. Thus we can approximate CCA objective (we will later check how good this approximation is) as

$$
\max_{\substack{\widetilde{U},\widetilde{V} \in \mathsf{St}(k,d) \\ R_u, R_v \in \mathbf{R}^{k \times k} \\ U = \widetilde{U}R_u;\ V = \widetilde{V}R_v}} \underbrace{\operatorname{trace}\left(U^T C_{XY} V\right)}_{\widetilde{F}} + \underbrace{\operatorname{trace}\left(\widetilde{U}^T C_X \widetilde{U}\right) + \operatorname{trace}\left(\widetilde{V}^T C_Y \widetilde{V}\right)}_{\widetilde{F}_{\text{pca}}} \quad \text{s.t.} \quad \begin{matrix} U^T C_X U = I_k \\ V^T C_Y V = I_k \end{matrix}
$$

(3)

Here, $\mathsf{St}(k,d)$ denotes the manifold consisting of $d \times k$ (with $k \leq d$) column orthonormal matrices, i.e., $\mathsf{St}(k,d) = \left\{X \in \mathbf{R}^{d \times k} | X^T X = I_k\right\}$. Observe that in (3), we approximate the optimal $U$ and $V$ as a linear combination of $\widetilde{U}$ and $\widetilde{V}$ respectively. *Thus, the aforementioned PCA solution can act as a feasible initial solution for (3).*

As the choice of $R_u$ and $R_v$ is arbitrary, we can further reparameterize these matrices by constraining them to be full rank (of rank $k$) and using the RQ decomposition Golub and Reinsch [1971] which gives us the following reformulation.

---

**A Reformulation for CCA**

$$
\max_{\substack{\widetilde{U},\widetilde{V},S_u,S_v,Q_u,Q_v \\ U = \widetilde{U}S_u Q_u;\ V = \widetilde{V}S_v Q_v}} \underbrace{\operatorname{trace}\left(U^T C_{XY} V\right)}_{\widetilde{F}} + \underbrace{\operatorname{trace}\left(\widetilde{U}^T C_X \widetilde{U}\right) + \operatorname{trace}\left(\widetilde{V}^T C_Y \widetilde{V}\right)}_{\widetilde{F}_{\text{pca}}}
$$
(4a)

subject to $\quad U^T C_X U = I_k$

$\qquad\qquad V^T C_Y V = I_k$ (4b)

$\qquad\qquad \widetilde{U}, \widetilde{V} \in \mathsf{St}(k,d);\ Q_u, Q_v \in \mathsf{SO}(k)$

$\qquad\qquad S_u, S_v$ is upper triangular

---

Here, $\mathsf{SO}(k)$ is the space of $k \times k$ special orthogonal matrices, i.e., $\mathsf{SO}(k) = \left\{X \in \mathbf{R}^{k \times k} | X^T X = I_k; \det(X) = 1\right\}$. Before evaluating how good the aforementioned approximation is, we first point out some useful properties of the reformulation (4): **(a)** in the reparametrization of $U$ and $V$, all components are structured, hence, the search space becomes a subset of $\mathbf{R}^{k \times k}$ **(b)** we can essentially initialize with a PCA solution and then try to optimize (4) via some scheme.

**Why (4) helps?** First, we note that CCA seeks to maximize the total correlation under the constraint that different components are decorrelated. One difficulty in the optimization is to ensure decorrelation, which leads to a higher complexity in existing streaming CCA algorithms. On the contrary, in (4), we separate (1) into finding the PCs, $\widetilde{U}, \widetilde{V}$ (by adding the variance maximization terms) and finding the linear combination ($S_u Q_u$ and $S_v Q_v$) of the principal directions. After optimizing for these variables, the whitening constraints are, up to a rescaling, automatically satisfied. Here, we can (almost) utilize an efficient off-the-shelf streaming PCA algorithm. We will defer describing the specific details of the individual steps until the next sub-section. First, we will show why substituting (1) with (4) is sensible under some assumptions.

**Why the solution of the reformulation makes sense?** We start by stating some mild assumptions needed for the analysis. **Assumptions: (a)** The random variables $\mathbf{X} \sim \mathcal{N}(\mathbf{0}, \Sigma_x)$ and $\mathbf{Y} \sim \mathcal{N}(\mathbf{0}, \Sigma_y)$ with covariance $\Sigma_x \preceq cI_d$ and covariance $\Sigma_y \preceq cI_d$ for some $c > 0$. **(b)** The samples $X$ and $Y$ drawn from $\mathcal{X}$ and $\mathcal{Y}$ respectively have zero mean. **(c)** For a given $k \leq d$, $\Sigma_x, \Sigma_y$ have non-zero top-$k$ eigen values.

We show how the presented solution, assuming access to an effective numerical procedure, approximates the CCA problem presented in (1). We formally state the result in the following theorem with a sketch of proof (appendix includes the full proof) by first stating the following proposition.

**Definition 1.** *A random variable $\mathbf{X}$ is called sub-Gaussian if the norm given by $\|\mathbf{X}\|_\star :=$ $\inf\left\{d \geq 0 | \mathbf{E_X}\left[\exp\left(\operatorname{trace}(X^T X)/d^2\right)\right] \leq 2\right\}$ is finite. Let $U \in \mathbf{R}^{d \times k}$, then $\mathbf{X}U$ is sub-Gaussian Vershynin [2017].*

**Proposition 1** (Reiß et al. [2020]). *Let $\mathbf{X}$ be a random variable which follows a sub-Gaussian distribution. Let $\widehat{X}$ be the approximation of $X \in \mathbf{R}^{N \times d}$ (samples drawn from $\mathcal{X}$) with the top-$k$*

*principal vectors. Let $\widetilde{C}_X$ be the covariance of $\widehat{X}$. Also, assume that $\lambda_i$ is the $i^{th}$ eigen value of $C_X$ for $i = 1, \cdots, d-1$ and $\lambda_i \geq \lambda_{i+1}$ for all $i$. Then, the PCA reconstruction error, denoted by $\mathcal{E}_k = \mathbf{E}_\mathbf{X}\|X - \widehat{X}\|$ (in the Frobenius norm sense) can be upper bounded as follows*

$$\mathcal{E}_k \leq \min\left( \sqrt{2k}\|\Delta\|_2, \frac{2\|\Delta\|_2^2}{\lambda_k - \lambda_{k+1}} \right), \quad \Delta = C_X - \widetilde{C}_X.$$

The aforementioned proposition suggests that the error between the data matrix $X$ and the reconstructed data matrix $\widehat{X}$ using the top-$k$ principal vectors is bounded.

Recall from (1) and (4) that the optimal value of the true and approximated CCA objective is denoted by $F$ and $\widetilde{F}$ respectively. The following theorem states that we can bound the error, $E = \|F - \widetilde{F}\|$ (proof in the appendix). In other words, if we start from PCA solution and can successfully optimize (4) without leaving the feasible set, we will obtain a good solution.

**Theorem 1.** *Using the hypothesis and assumptions above, the approximation error $E = \|F - \widetilde{F}\|$ as a function of $N$ is bounded and goes to zero as $N \to \infty$ while the whitening constraints in equation 4b are satisfied.*

*Sketch of the Proof.* Let $U^*$ and $V^*$ be the true solution of CCA, i.e., of (1). Let $U = \widetilde{U}S_uQ_u, V = \widetilde{V}S_vQ_v$ be the solution of (4), with $\widetilde{U}, \widetilde{V}$ be the PCA solutions of $X$ and $Y$ respectively. Let $\widehat{X} = X\widetilde{U}\widetilde{U}^T$ and $\widehat{Y} = Y\widetilde{V}\widetilde{V}^T$ be the reconstruction of $X$ and $Y$ using principal vectors. Let $S_uQ_u = \widetilde{U}^TU^*$ and $S_vQ_v = \widetilde{V}^TV^*$. Then we can write $\widetilde{F} = \text{trace}\left(U^TC_{XY}V\right)$ $= \text{trace}\left(\frac{1}{N}\left(\widehat{X}U^*\right)^T \widehat{Y}V^*\right)$. Similarly we can write $F = \text{trace}\left(\frac{1}{N}\left(XU^*\right)^T YV^*\right)$. As $\widehat{X}$ and $\widehat{Y}$ are the approximation of $X$ and $Y$ respectively using the principal vectors, we use Prop. 1 to bound the error $\|F - \widetilde{F}\|$. Now observe that $\widehat{X}U$ can be rewritten into $X\widetilde{U}\widetilde{U}^TU$ (similar for $\widehat{Y}V$). Thus, as long as the solution $S_uQ_u$ and $S_vQ_v$ respectively well-approximate $\widetilde{U}^TU$ and $\widetilde{V}^TV$, $\widetilde{F}$ is a good approximation of $F$. $\qquad\square$

Now, the only unresolved issue is an optimization scheme for equation 4a that keeps the constraints in equation 4b satisfied by leveraging the geometry of the structured solution space.

### 2.1 How to numerically optimize (4a) satisfying constraints in (4b)?

**Overview.** We now describe how to maximize the formulation in (4a)–(4b) with respect to $\widetilde{U}, \widetilde{V}, Q_u, Q_v, S_u$ and $S_v$. We will first compute top-$k$ principal vectors to get $\widetilde{U}$ and $\widetilde{V}$. Then, we will use a gradient update rule to solve for $Q_u, Q_v, S_u$ and $S_v$ to improve the objective. Since all these matrices are "structured", care must be taken to ensure that the matrices *remain on their respective manifolds* – which is where the geometry of the manifolds will offer desirable properties. We re-purpose a Riemannian stochastic gradient descent (RSGD) to achieve this task, so call our algorithm *RSG+*. Of course, more sophisticated Riemannian optimization techniques can be substituted in. For instance, different Riemannian optimization methods are available in Absil et al. [2007] and optimization schemes for many manifolds are offered in PyManOpt Boumal et al. [2014].

The algorithm block is in Algorithm 1. Recall that $\widetilde{F}_{\text{pca}} = \text{trace}\left(U^TC_XU\right) + \text{trace}\left(V^TC_YV\right)$ is the contribution from the principal directions which we used to ensure the "whitening constraint". Moreover, $\widetilde{F} = \text{trace}\left(U^TC_{XY}V\right)$ is the contribution from the canonical correlation directions (*note that we use the subscript 'cca' for making CCA objective explicit*). The algorithm consists of four main blocks denoted by different colors, namely **(a)** the Red block deals with gradient calculation of the objective function where we calculate the top-$k$ principal vectors (denoted by $\widetilde{F}_{\text{pca}}$) with respect to $\widetilde{U}, \widetilde{V}$; **(b)** the Green block describes calculation of the gradient corresponding to the canonical directions (denoted by $\widetilde{F}$) with respect to $\widetilde{U}, \widetilde{V}, S_u, S_v, Q_u$ and $Q_v$; **(c)** the Gray block combines the gradient computation from both $\widetilde{F}_{\text{pca}}$ and $\widetilde{F}$ with respect to unknowns $\widetilde{U}, \widetilde{V}, S_u, S_v, Q_u$ and $Q_v$; and finally **(d)** the Blue block performs a batch update of the canonical directions $\widetilde{F}$ using Riemannian gradient updates.

**Gradient calculations.** The gradient update for $\widetilde{U}, \widetilde{V}$ is divided into two parts **(a)** The (Red block) gradient updates the "principal" directions (denoted by $\nabla_{\widetilde{U}} \widetilde{F}_{\text{pca}}$ and $\nabla_{\widetilde{V}} \widetilde{F}_{\text{pca}}$), which is specifically designed to satisfy the *whitening constraint*. This requires updating the principal subspaces, so, the gradient descent needs to proceed on the manifold of $k$-dimensional subspaces of $\mathbf{R}^d$, i.e., on the Grassmannian $\mathsf{Gr}(k, d)$. **(b)** The (green block) gradient from the objective function in (4), is denoted by $\nabla_{\widetilde{U}} \widetilde{F}$ and $\nabla_{\widetilde{V}} \widetilde{F}$. In order to ensure that the Riemannian gradient update for $\widetilde{U}$ and $\widetilde{V}$ stays on the manifold $\mathsf{St}(k, d)$, we need to make sure that the gradients, i.e., $\nabla_{\widetilde{U}} \widetilde{F}$ and $\nabla_{\widetilde{V}} \widetilde{F}$ lies in the tangent space of $\mathsf{St}(k, d)$. To do so, we need to first calculate the Euclidean gradient and then project on to the tangent space of $\mathsf{St}(k, d)$.

The gradient updates for $Q_u, Q_v, S_u, S_v$ are given in the green block, denoted by $\nabla_{Q_u} \widetilde{F}$, $\nabla_{Q_v} \widetilde{F}$, $\nabla_{S_u} \widetilde{F}$ and $\nabla_{S_v} \widetilde{F}$. Note that unlike the previous step, this gradient only has components from canonical correlation calculation. As before, this step requires first computing the Euclidean gradient and then projecting on to the tangent space of the underlying Riemannian manifolds involved, i.e., $\mathsf{SO}(k)$ and the space of upper triangular matrices.

Finally, we get the gradient to update the canonical directions by combining the gradients which is shown in the gray block. With these gradients we can perform a batch update as shown in the blue block. A schematic diagram is given in Fig. 1.

Using results presented next in Propositions 2–3, this scheme can be shown (under some assumptions) to approximately optimize the CCA objective in (1).

We can now move to the convergence properties of the algorithm. We present two results stating the asymptotic proof of convergence for top-$k$ principal vectors and canonical directions in the algorithm.

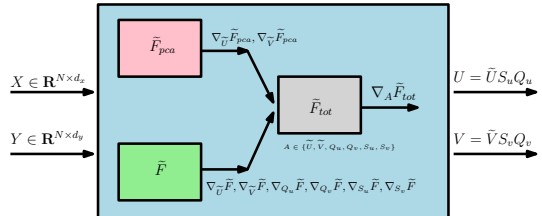

Figure 1: Schematic diagram of the proposed CCA algorithm, here $\widetilde{F}_{\text{tot}} = \widetilde{F} + \widetilde{F}_{\text{pca}}$, where $\widetilde{F}$ is the approximated objective value for CCA (as in (4))

**Proposition 2** (Chakraborty et al. [2020]). *(Asymptotically) If the samples, $X$, are drawn from a Gaussian distribution, then the gradient update rule presented in Step 5 in Algorithm 1 returns an orthonormal basis – the top-k principal vectors of the covariance matrix $C_X$.*

**Proposition 3.** *(Bonnabel [2013]) Consider a connected Riemannian manifold $\mathcal{M}$ with injectivity radius bounded from below by $I > 0$. Assume that the sequence of step sizes $(\gamma_l)$ satisfy the condition (a) $\sum \gamma_l^2 < \infty$ (b) $\sum \gamma_l = \infty$. Suppose $\{A_l\}$ lie in a compact set $K \subset \mathcal{M}$. We also suppose that $\exists D > 0$ such that, $g_{A_l} \left( \nabla_{A_l} \widetilde{F}, \nabla_{A_l} \widetilde{F} \right) \leq D$. Then $\nabla_{A_l} \widetilde{F} \to 0$ and $l \to \infty$.*

Notice that in our problem, the injectivity radius bound in Proposition 3 is satisfied as "$I$" for $\mathsf{Gr}(p, n)$, $\mathsf{St}(p, n)$ or $\mathsf{SO}(p)$ is $\pi/2\sqrt{2}, \pi/2\sqrt{2}, \pi/2$ respectively. So, in order to apply Proposition 3, we need to guarantee the step sizes satisfy the aforementioned condition. One example of the step sizes that satisfies the property is $\gamma_l = \frac{1}{l+1}$.

### 2.2 Convergence rate and complexity of the RSG+ algorithm

In this section, we describe the convergence rate and complexity of the algorithm proposed in Algorithm 1. Observe that the key component of Algorithm 1 is a Riemannian gradient update. Let $A_t$ be the generic entity needed to be updated in the algorithm using the Riemannian gradient update $A_{t+1} = \mathsf{Exp}_{A_t} \left( -\gamma_t \nabla_{A_t} \widetilde{F} \right)$, where $\gamma_t$ is the step size at time step $t$. Also assume $\{A_t\} \subset \mathcal{M}$ for a Riemannian manifold $\mathcal{M}$. The following proposition states that under certain assumptions, the Riemannian gradient update has a convergence rate of $O\left(\frac{1}{t}\right)$.

**Proposition 4.** *(Nemirovski et al. [2009], Bécigneul and Ganea [2018]) Let $\{A_t\}$ lie inside a geodesic ball of radius less than the minimum of the injectivity radius and the strong convexity radius of $\mathcal{M}$. Assume $\mathcal{M}$ to be a geodesically complete Riemannian manifold with sectional curvature lower bounded by $\kappa \leq 0$. Moreover, assume that the step size $\{\gamma_t\}$ diverges and the squared step size converges. Then, the Riemannian gradient descent update given by $A_{t+1} = \mathsf{Exp}_{A_t} \left( -\gamma_t \nabla_{A_t} \widetilde{F} \right)$*

---

**Algorithm 1:** Riemannian SGD based algorithm (RSG+) to compute canonical directions

1 **Input:** $X \in \mathbf{R}^{N \times d_x}, Y \in \mathbf{R}^{N \times d_y}, k > 0$

2 **Output:** $U \in \mathbf{R}^{d_x \times k}, V \in \mathbf{R}^{d_y \times k}$

3 Initialize $\widetilde{U}, \widetilde{V}, Q_u, Q_v, S_u, S_v$

4 Partition data $X, Y$ into batches of size $B$. Let $j^{th}$ batch be denoted by $X_j$ and $Y_j$

5 **for** $j \in \left\{ 1, \cdots, \lfloor \frac{N}{B} \rfloor \right\}$ **do**

6

> **Gradient for top-$k$ principal vectors**: calculating $\nabla_{\widetilde{U}} \widetilde{F}_{\text{pca}}, \nabla_{\widetilde{V}} \widetilde{F}_{\text{pca}}$
>
> 1. Partition $X_j$ ($Y_j$) into $L$ ($L = \lfloor \frac{B}{k} \rfloor$) blocks of size $d_x \times k$ ($d_y \times k$);
> 2. Let the $l^{th}$ block be denoted by $Z_l^x$ ($Z_l^y$);
> 3. Orthogonalize each block and let the orthogonalized block be denoted by $\hat{Z}_l^x$ ($\hat{Z}_l^y$);
> 4. Let the subspace spanned by each $\hat{Z}_l^x$ (and $\hat{Z}_l^y$) be $\hat{\mathcal{Z}}_l^x \in \mathsf{Gr}(k, d_x)$ (and $\hat{\mathcal{Z}}_l^y \in \mathsf{Gr}(k, d_y)$);
>
> $$\nabla_{\widetilde{U}} \widetilde{F}_{\text{pca}} = - \sum_l \mathsf{Exp}_{\widetilde{U}}^{-1} \left( \hat{\mathcal{Z}}_l^x \right) \quad \nabla_{\widetilde{V}} \widetilde{F}_{\text{pca}} = - \sum_l \mathsf{Exp}_{\widetilde{V}}^{-1} \left( \hat{\mathcal{Z}}_l^y \right) \tag{5}$$

7

> **Gradient from equation 4**: calculating $\nabla_{\widetilde{U}} \widetilde{F}, \nabla_{\widetilde{V}} \widetilde{F}, \nabla_{Q_u} \widetilde{F}, \nabla_{Q_v} \widetilde{F}, \nabla_{S_u} \widetilde{F}, \nabla_{S_v} \widetilde{F}$
>
> $\nabla_{\widetilde{U}} \widetilde{F} = \frac{\partial \widetilde{F}}{\partial \widetilde{U}} - \widetilde{U} \frac{\partial \widetilde{F}}{\partial \widetilde{U}}^T \widetilde{U}$  $\quad \nabla_{\widetilde{V}} \widetilde{F} = \frac{\partial \widetilde{F}}{\partial \widetilde{V}} - \widetilde{V} \frac{\partial \widetilde{F}}{\partial \widetilde{V}}^T \widetilde{V}$
>
> $\nabla_{Q_u} \widetilde{F} = \frac{\partial \widetilde{F}}{\partial Q_u} - \frac{\partial \widetilde{F}}{\partial Q_u}^T$  $\quad \nabla_{Q_v} \widetilde{F} = \frac{\partial \widetilde{F}}{\partial Q_v} - \frac{\partial \widetilde{F}}{\partial Q_v}^T$
>
> $\nabla_{S_u} \widetilde{F} = \mathsf{Upper} \left( \frac{\partial \widetilde{F}}{\partial S_u} \right)$  $\quad \nabla_{S_v} \widetilde{F} = \mathsf{Upper} \left( \frac{\partial \widetilde{F}}{\partial S_v} \right)$
>
> Here, Upper returns the upper triangular matrix of the input matrix and $\frac{\partial \widetilde{F}}{\partial \widetilde{U}}, \frac{\partial \widetilde{F}}{\partial \widetilde{V}}, \frac{\partial \widetilde{F}}{\partial Q_u}, \frac{\partial \widetilde{F}}{\partial Q_v}, \frac{\partial \widetilde{F}}{\partial S_u}, \frac{\partial \widetilde{F}}{\partial S_v}$ give the Euclidean gradients, which are provided in appendix.

8

> **Gradient to update canonical directions**
>
> $\nabla_{\widetilde{U}} \widetilde{F}_{\text{tot}} = \nabla_{\widetilde{U}} \widetilde{F}_{\text{pca}} + \nabla_{\widetilde{U}} \widetilde{F}$  $\qquad \nabla_{\widetilde{V}} \widetilde{F}_{\text{tot}} = \nabla_{\widetilde{V}} \widetilde{F}_{\text{pca}} + \nabla_{\widetilde{V}} \widetilde{F}$;
>
> $\nabla_X \widetilde{F}_{\text{tot}} = \nabla_X \widetilde{F}$ where, $X$ is a generic entity: $X \in \{Q_u, Q_v, S_u, S_v\}$;

9

> **Batch update of canonical directions**
>
> $A = \mathsf{Exp}_A \left( -\gamma_j \nabla_A \widetilde{F}_{\text{tot}} \right)$ where, $A$ is a generic entity: $A \in \{\widetilde{U}, \widetilde{V}, Q_u, Q_v, S_u, S_v\}$;

10 **end for**

11 $U = \widetilde{U} Q_u S_u$ and $V = \widetilde{V} Q_v S_v$;

---

*with a bounded $\nabla_{A_t} \widetilde{F}$, i.e., $\|\nabla_{A_t} \widetilde{F}\| \leq C < \infty$ for some $C \geq 0$, converges in the rate of $O\left(\frac{1}{t}\right)$ with the number of iterates bounded by $\mathcal{O}(N + D/\epsilon^2)$, for some tolerance $\epsilon > 0$ and for the Lipschitz bound $D$ of the objective function $\widetilde{F}$.*

For this result to be applicable, we need the CCA objective function to be geodesically convex as a function of $U$ and $V$ (proof in the appendix). All Riemannian manifolds we needed, i.e., $\mathsf{Gr}(k, d)$, $\mathsf{St}(k, d)$ and $\mathsf{SO}(k)$ are geodesically complete, and these manifolds have non-negative sectional curvatures, i.e., lower bounded by $\kappa = 0$. Moreover the minimum of convexity and injectivity radius for $\mathsf{Gr}(k, d)$, $\mathsf{St}(k, d)$ and $\mathsf{SO}(k)$ are $\pi/2\sqrt{2}$. Now, as long as the Riemannian updates lie inside the geodesic ball of radius less than $\pi/2\sqrt{2}$, the convergence rate for RGD applies in our setting.

**Running time.** To evaluate time complexity, we must look at the main compute-heavy steps needed. The basic modules are $\mathsf{Exp}$ and $\mathsf{Exp}^{-1}$ maps for $\mathsf{St}(k, d)$, $\mathsf{Gr}(k, d)$ and $\mathsf{SO}(k)$ manifolds (see Table 1 in appendix for a detailed specification of these maps). Observe that the complexity of these modules is influenced by the complexity of $\mathsf{svd}$ needed for the $\mathsf{Exp}$ map for the $\mathsf{St}$ and $\mathsf{Gr}$ manifolds. Our algorithm involves structured matrices of size $d \times k$ and $k \times k$, so any matrix operation should not exceed a cost of $O(\max(d^2 k, k^3))$, since in general $d \gg k$. Specifically, the most expensive calculation is SVD of matrices of size $d \times k$, which is $O(d^2 k)$, see Golub and Reinsch [1971]. All other calculations are dominated by this term.

Table 1: Wall-clock runtime of one pass through the data of our RSG+ and MSG on MNIST, Mediamill and CIFAR (average of 5 runs).

| Time (s) | MNIST | | | Mediamill | | | CIFAR | | |
|---|---|---|---|---|---|---|---|---|---|
| | $k=1$ | $k=2$ | $k=4$ | $k=1$ | $k=2$ | $k=4$ | $k=1$ | $k=2$ | $k=4$ |
| RSG+ (Ours) | 4.16 | 4.24 | 4.71 | 1.89 | 1.60 | 1.44 | 14.80 | 17.22 | 22.10 |
| MSG | 35.32 | 42.09 | 49.17 | 11.59 | 14.21 | 17.34 | 80.21 | 100.80 | 106.55 |

## 3 Experiments

We first evaluate RSG+ for extracting top-$k$ canonical components on three benchmark datasets and show that it performs favorably compared with Arora et al. [2017]. Then, we show that RSG+ also fits into feature learning in DeepCCA Andrew et al. [2013], and can scale to large feature dimensions where the non-stochastic method fails. Finally, we show that RSG+ can be used to improve fairness of deep neural networks without full access to labels of protected attributes during training.

### 3.1 CCA on Fixed Datasets

**Datasets and baseline.** We conduct experiments on three benchmark datasets (MNIST LeCun et al. [2010], Mediamill Snoek et al. [2006] and CIFAR-10 Krizhevsky [2009]) to evaluate the performance of RSG+ to extract top-$k$ canonical components. To our knowledge, Arora et al. [2017] is the only previous work which stochastically optimizes the population objective in a streaming fashion and can extract top-$k$ components, so we compare our RSG+ with the matrix stochastic gradient (MSG) method proposed in Arora et al. [2017] (note: there are two methods proposed in Arora et al. [2017] and we choose MSG because it performs better in the experiments in Arora et al. [2017]). The details regarding the three datasets and how we process them are as follows:

**MNIST.** LeCun et al. [2010]: MNIST contains grey-scale images of size $28 \times 28$. We use its full training set containing 60K images. Every image is split into left/right half, which are used as the two views. **Mediamill.** Snoek et al. [2006]: Mediamill contains around 25.8K paired features of videos and corresponding commentary of dimension $120, 101$ respectively. **CIFAR-10.** Krizhevsky [2009]: CIFAR-10 contains 60K $32 \times 32$ color images. Like MNIST, we split the images into left/right half and use them as two views.

**Evaluation metric.** We choose to use Proportion of Correlations Captured (PCC) which is widely used Ma et al. [2015], Ge et al. [2016], partly due to its efficiency, especially for relatively large datasets. Let $\hat{U} \in R^{d_x \times k}, \hat{V} \in R^{d_y \times k}$ denote the estimated subspaces returned by RSG+, and $U^* \in R^{d_x \times k}, V^* \in R^{d_y \times k}$ denote the true canonical subspaces (all for top-$k$). The PCC is defined as $\text{PCC} = \frac{\text{TCC}(X\hat{U}, Y\hat{V})}{\text{TCC}(XU^*, YV^*)}$, where TCC is the sum of canonical correlations between two matrices.

**Performance.** We run our algorithm with step sizes chosen from {1, 0.1, 0.01, 0.001, 0.0001, 0.00001}. The performance in terms of PCC as a function of the number of seen samples (shown in a streaming manner) are shown in Fig. 2, and our RSG+ achieves around $10\times$ runtime improvement over MSG (see Table 1). Our RSG+ captures more correlation than MSG Arora et al. [2017] while being $5 - 10$ times faster. One case where our RSG+ underperforms Arora et al. [2017] is when the top-$k$ eigenvalues are dominated by the top-$l$ eigenvalues with $l < k$ (Fig. 2b): on Mediamill dataset, the top-4 eigenvalues of the covariance matrix in view 1 are: $8.61, 2.99, 1.15, 0.37$. The first eigenvalue is dominantly large compared to the rest and our RSG+ performs better for $k = 1$ and worse than Arora et al. [2017] for $k = 2, 4$. Runtime of RSG+ for different data dimensions (set $d_x = d_y = d$) and number of total samples (from a joint Gaussian distribution) is in the appendix.

### 3.2 CCA for Deep Feature Learning

**Background and motivation.** A deep neural network (DNN) extension of CCA was proposed by Andrew et al. [2013] and has become popular in multi-view representation learning tasks. The idea is to learn a deep neural network as the mapping from original data space to a latent space where the canonical correlations are maximized. We refer the reader to Andrew et al. [2013] for details of the task. Since deep neural networks are usually trained using SGD on mini-batches, this requires

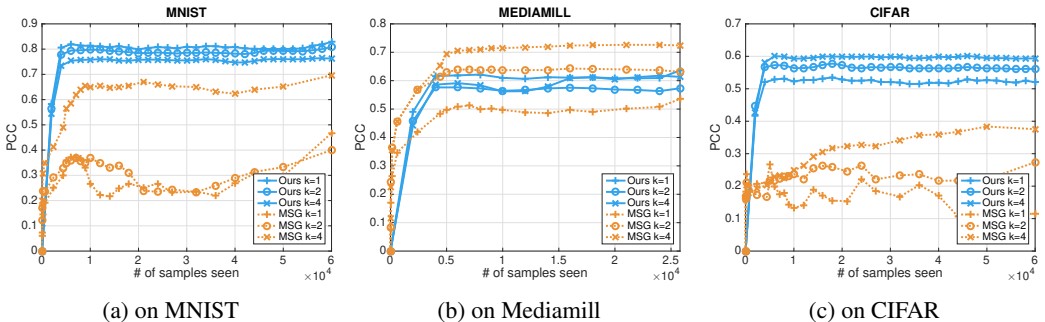

| (a) on MNIST | (b) on Mediamill | (c) on CIFAR |

Figure 2: Performance on three datasets in terms of PCC as a function of # of seen samples.

obtaining an estimate of the CCA objective at every iteration in a streaming fashion, thus our RSG+ can be a natural fit. We conduct experiments on a noisy version of MNIST dataset to evaluate RSG+.

**Dataset.** We follow Wang et al. [2015a] to construct a noisy version of MNIST: View 1 is a randomly sampled image which is first rescaled to $[0, 1]$ and then rotated by a random angle from $[-\frac{\pi}{4}, \frac{\pi}{4}]$. View 2 is randomly sampled from the same class as view 1. Then we add independent uniform noise from $[0, 1]$ to each pixel. Finally the image is truncated into $[0, 1]$ to form view 2.

Table 2: Results of feature learning on MNIST. N/A means fails to yield a result on our hardware.

| Accuracy(%) | $d = 100$ | $d = 500$ | $d = 1000$ |
|---|---|---|---|
| DeepCCA | 80.57 | $N/A$ | $N/A$ |
| Ours | 79.79 | 84.09 | 86.39 |

**Implementation details.** We use a simple 2-layer MLP with ReLU nonlinearity, where the hidden dimension in the middle is $512$ and the output feature dimension is $d \in \{100, 500, 1000\}$. After the network is trained on the CCA objective, we use a linear Support Vector Machine (SVM) to measure classification accuracy on output latent features. Andrew et al. [2013] uses the closed form CCA objective on the current batch directly, which costs $O(d^3)$ memory and time for every iteration.

**Performance.** Table 2 shows that we get similar performance when $d = 100$ and can scale to large latent dimensions $d = 1000$ while the batch method Andrew et al. [2013] encounters numerical difficulty on our GPU resources and the Pytorch Paszke et al. [2019] platform in performing an eigen-decomposition of a $d \times d$ matrix when $d = 500$, and becomes difficult if $d$ is larger than $1000$.

## 3.3 CCA for Fairness applications

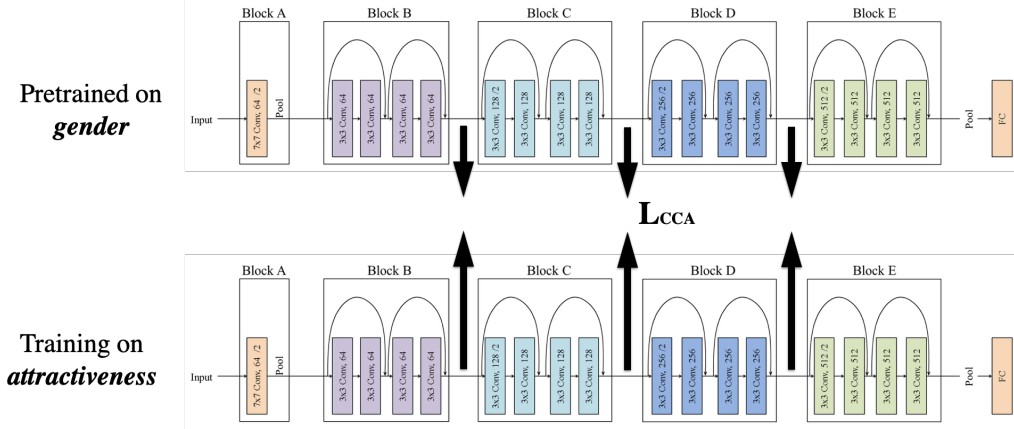

Figure 3: Training architecture for fairness experiment. The model above is the pretrained model and the model below is being trained. Use of CCA allows the two network architectures to be different.

**Background and motivation.** Fairness is becoming an important issue to consider in the design of learning algorithms. A common strategy to make an algorithm fair is to remove the influence of

one/more protected attributes when training the models, see Lokhande et al. [2020]. Most methods assume that the labels of protected attributes are known during training but this may not always be possible. CCA enables considering a slightly different setting, where we may not have per-sample protected attributes which may be sensitive or hard to obtain for third-parties Price and Cohen [2019]. On the other hand, we assume that a model pre-trained to predict the protected attribute labels is provided. For example, if the protected attribute is gender, we only assume that a good classifier which is trained to predict gender from the samples is available rather than sample-wise gender values themselves. We next demonstrate that fairness of the model, using standard measures, can be improved via constraints on correlation values from CCA.

**Dataset.** CelebA Wang et al. [2015b] consists of 200K celebrity face images from the internet. There are up to 40 labels, each of which is binary-valued. Here, we follow Lokhande et al. [2020] to focus on the *attactiveness* attribute (which we want to train a classifier to predict) and the *gender* is treated as "protected" since it may lead to an unfair classifier according to Lokhande et al. [2020].

**Method.** Our strategy is inspired by Morcos et al. [2018] which showed that canonical correlations can reveal the similarity in neural networks: when two networks (same architecture) are trained using different labels/schemes for example, canonical correlations can indicate how similar their features are. Our observation is the following. Consider a classifier that is trained on gender (the pro-

Table 3: Fairness results on CelebA. We applied CCA on three different layers in Resnet-18 respectively. See appendix for positions of conv $0, 1, 2$. "Ours-conv[0,1]-conv[1,2]" means stacking features from different layers to form hypercolumn features Hariharan et al. [2015], which shows that our approach allows two networks to have different shape/size.

|  | Accuracy(%) | DEO(%) | DDP(%) |
|---|---|---|---|
| Unconstrained | 76.3 | 22.3 | 4.8 |
| Ours-conv0 | 76.5 | 17.4 | **1.4** |
| Ours-conv1 | **77.7** | **15.3** | 3.2 |
| Ours-conv2 | 75.9 | 22.0 | 2.8 |
| Ours-conv[0,1]-conv[1,2] | 76.0 | 22.1 | 3.9 |

tected attribute), and another classifier that is trained on *attractiveness*, if the features extracted by the latter model share a high similarity with the one trained to predict gender, then it is more likely that the latter model is influenced by features in the image pertinent to gender, which will lead to an unfairly biased trained model. We show that by imposing a loss on the canonical correlation between the network being trained (but we lack per-sample protected attribute information) and a well trained classifier pre-trained on the protected attributes, we can obtain a more fair model. This may enable training fairer models in settings which would otherwise be difficult. The training architecture is shown in Fig. 3.

**Implementation details.** To simulate the case where we only have a pretrained network on protected attributes, we train a Resnet-18 He et al. [2016] on the *gender* attribute, and when we train the classifier to predict *attractiveness*, we add a loss using the canonical correlations between these two networks on intermediate layers: $L_{\text{total}} = L_{\text{cross-entropy}} + L_{\text{CCA}}$ where the first term is the standard cross entropy term and the second term is the canonical correlation. See appendix for more details of training/evaluation.

**Results.** We choose two commonly used error metrics for fairness: difference in Equality of Opportunity Hardt et al. [2016] (DEO), and difference in Demographic Parity Yao and Huang [2017] (DDP). We conduct experiments by applying the canonical correlation loss on three different layers in Resnet-18. In Table 3, we can see that applying canonical correlation loss generally improves the DEO and DDP metrics (lower is better) over the standard model (trained using cross entropy loss only). Specifically, applying the loss on early layers like conv0 and conv1 gets better performance than applying at a relatively late layer like conv2. Another promising aspect of our approach is that is can easily handle the case where the protected attribute is a continuous variable (as long as a well trained regression network on the protected attribute is given) while other methods like Lokhande et al. [2020], Zhang et al. [2018] need to first discretize the variable and then enforce constraints which can be much more involved.

**Limitations.** Our current implementation has difficulty to scale beyond $d = 10^5$ data dimension and this may be desirable for large scale DNNs. Exploring sparsity may be one way to solve the problem and will be enabled by additional developments in modern toolboxes.

# 4 Related Work

**Stochastic CCA:** There has been much interest in designing scalable and provable algorithms for CCA: Ma et al. [2015] proposed the first stochastic algorithm for CCA, where local convergence is proven for the non-stochastic version. Wang et al. [2016] designed an algorithm which uses alternating SVRG combined with shift-and-invert pre-conditioning, with global convergence properties. These stochastic methods, and Ge et al. [2016] Allen-Zhu and Li [2016], which reduce the CCA problem to a generalized eigenvalue problem and solve it via an efficient power method, all belong to the class of methods that seeks to to solve the empirical CCA problem. It can be seen as an ERM approximation of the original population objective, which requires solving numerically the empirical CCA objective on a fixed data set. These methods usually assume access to the full dataset at the outset, which may not be suitable for some practical applications where data is presented in a streaming manner. Recently, there appears to be an interest in considering the population CCA problem Arora et al. [2017] Gao et al. [2019]. The main difficulty in the population setting is that we have limited knowledge about the objective unless we know the distribution of $\mathbf{X}$ and $\mathbf{Y}$. Arora et al. [2017] handles this problem by deriving an estimation of the gradient of the population objective whose error can be properly bounded so that applying proximal gradient to a convex relaxed objective will provably converge. Gao et al. [2019] provides a tightened analysis of the time complexity of the algorithm in Wang et al. [2016], and provides sample complexity for certain distributions. The problem we study is similar to the one in Arora et al. [2017], Gao et al. [2019]: to optimize the population objective of CCA in a streaming fashion.

**Riemannian Optimization:** Riemannian optimization is a generalization of standard Euclidean optimization methods to smooth manifolds, which takes the following form: given $f : \mathcal{M} \to \mathbf{R}$, solve $\min_{x \in \mathcal{M}} f(x)$, where $\mathcal{M}$ is a Riemannian manifold. Advantages often include efficient numerical procedures for certain classes of constrained optimization problems. Applications include matrix and tensor factorization Ishteva et al. [2011], Tan et al. [2014], PCA Edelman et al. [1998], CCA Yger et al. [2012], and so on. We remark that Yger et al. [2012] also describes CCA formulation by rewriting it as a Riemannian optimization on the Stiefel manifold. In our work, we further explore the benefits of the Riemannian optimization toolkit, decomposing the linear space spanned by canonical vectors into products of several matrices which lie in several different Riemannian manifolds.

# 5 Conclusions

In this work, we presented a stochastic approach (RSG+) for the CCA model based on the observation that the solution of CCA can be decomposed into a product of matrices which lie on certain structured spaces. This affords specialized numerical schemes and makes the optimization more efficient. The optimization is based on Riemannian stochastic gradient descent and we provide a proof for its $O(\frac{1}{t})$ convergence rate with the number of iterates upper bounded, with $O(d^2 k)$ time complexity per iteration. In experimental evaluations, we find that our RSG+ behaves favorably relative to the baseline stochastic CCA method in capturing the correlation in the datasets. We also show the use of RSG+ in the DeepCCA setting showing feasibility when scaling to large dimensions as well as in an interesting use case in training fair models.

Our full codebase is available for use at https://github.com/zihangm/riemannian-streaming-cca.

**Potential negative societal impacts.** Since CCA is a fundamental problem in statistical machine learning and not tied to specific applications, we do not see a negative societal impact of our proposed method. However, it is possible that CCA can be used to uncover relationships between measurements which can be used for undesirable purposes.

# 6 Acknowledgments

This work was supported by NIH RF1AG059312, RF1AG062336 and R01EB022883, and NSF CCF #1918211. We thank Vishnu Lokhande for help with setting up the fairness experiments.

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
