# An Online Riemannian PCA for Stochastic Canonical Correlation Analysis: Supplementary Material

**Zihang Meng**[1]* **Rudrasis Chakraborty**[2]* **Vikas Singh**[1]
[1]University of Wisconsin-Madison
[2]Butlr
zihangm@cs.wisc.edu, rudrasischa@gmail.com, vsingh@biostat.wisc.edu

## 1 Appendix

### 1.1 A brief review of relevant differential geometry concepts

To make the paper self-contained, we briefly review certain differential geometry concepts. We only include a condensed description – needed for our algorithm and analysis – and refer the interested reader to Boothby (1986) for a comprehensive and rigorous treatment of the topic.

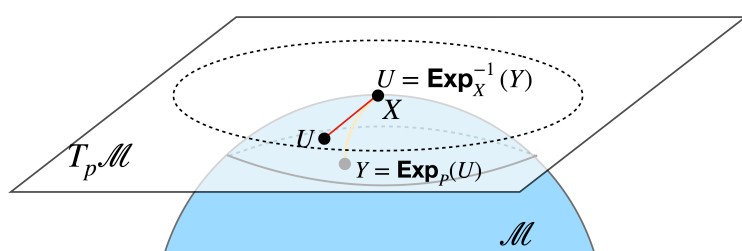

Figure 1: Schematic description of an exemplar manifold ($\mathcal{M}$) and the visual illustration of $\mathsf{Exp}$ and $\mathsf{Exp}^{-1}$ map.

**Riemannian Manifold:** A Riemannian manifold, $\mathcal{M}$, (of dimension $m$) is defined as a (smooth) topological space which is locally diffeomorphic to the Euclidean space $\mathbf{R}^m$. Additionally, $\mathcal{M}$ is equipped with a Riemannian metric $g$ which can be defined as

$$g_X : T_X\mathcal{M} \times T_X\mathcal{M} \to \mathbf{R},$$

where $T_X\mathcal{M}$ is the tangent space at $X$ of $\mathcal{M}$, see Fig. 1.

If $X \in \mathcal{M}$, the Riemannian Exponential map at $X$, denoted by $\mathsf{Exp}_X : T_X\mathcal{M} \to \mathcal{M}$ is defined as $\gamma(1)$ where $\gamma : [0,1] \to \mathcal{M}$. We can find $\gamma$ by solving the following differential equation:

$$\gamma(0) = X, (\forall t_0 \in [0,1]) \frac{d\gamma}{dt}\Big|_{t=t_0} = U.$$

In general $\mathsf{Exp}_X$ is not invertible but the inverse

$$\mathsf{Exp}_X^{-1} : \mathcal{U} \subset \mathcal{M} \to T_X\mathcal{M}$$

is defined only if $\mathcal{U} = \mathcal{B}_r(X)$, where $r$ is called the *injectivity radius* Boothby (1986) of $\mathcal{M}$. This concept will be useful to define the mechanics of gradient descent on the manifold.

In our reformulation, we made use of the following manifolds, specifically, when decomposing $U$ and $V$ into a product of several matrices.

---

*Equal contribution

35th Conference on Neural Information Processing Systems (NeurIPS 2021), virtual.

(a) $\mathsf{St}(p, n)$: the **Stiefel** manifold consists of $n \times p$ column orthonormal matrices

(b) $\mathsf{Gr}(p, n)$: the **Grassman** manifold consists of $p$-dimensional subspaces in $\mathbf{R}^n$

(c) $\mathsf{SO}(n)$, the manifold/group consists of $n \times n$ **special orthogonal matrices**, i.e., space of orthogonal matrices with determinant 1.

**Differential Geometry of $\mathsf{SO}(n)$:** $\mathsf{SO}(n)$ is a compact Riemannian manifold, hence by the Hopf-Rinow theorem, it is also a geodesically complete manifold Helgason (2001). Its geometry is well understood – we recall a few relevant concepts here and note that Helgason (2001) includes a more comprehensive treatment.

$\mathsf{SO}(n)$ has a Lie group structure and the corresponding Lie algebra, $\mathfrak{so}(n)$, is defined as,

$$\mathfrak{so}(n) = \{W \in \mathbf{R}^{n \times n} | W^T = -W\}.$$

In other words, $\mathfrak{so}(n)$ (the set of Left invariant vector fields with associated Lie bracket) is the set of $n \times n$ anti-symmetric matrices. The Lie bracket, $[,]$, operator on $\mathfrak{so}(n)$ is defined as the commutator, i.e.,

$$\text{for } U, V \in \mathfrak{so}(n), \quad [U, V] = UV - VU.$$

Now, we can define a Riemannian metric on $\mathsf{SO}(n)$ as follows:

$$\langle U, V \rangle_X = \text{trace}\left(U^T V\right), \quad \text{where}$$

$$U, V \in T_X(\mathsf{SO}(n)), X \in \mathsf{SO}(n).$$

It can be shown that this is a bi-invariant Riemannian metric. Under this bi-invariant metric, now we define the Riemannian exponential and inverse exponential map as follows. Let, $X, Y \in \mathsf{SO}(n)$, $U \in T_X(\mathsf{SO}(n))$. Then,

$$Exp_X^{-1}(Y) = X \log(X^T Y)$$

$$Exp_X(U) = X \exp(X^T U),$$

where, $\exp$, $\log$ are the matrix exponential and logarithm respectively.

**Differential Geometry of the Stiefel manifold:** The set of all full column rank $(n \times p)$ dimensional real matrices form a Stiefel manifold, $\mathsf{St}(p, n)$, where $n \geq p$.

A compact Stiefel manifold is the set of all column orthonormal real matrices. When $p < n$, $\mathsf{St}(p, n)$ can be identified with

$$\mathsf{SO}(n)/SO(n - p).$$

Note that, when we consider the quotient space, $\mathsf{SO}(n)/SO(n - p)$, we assume that $\mathsf{SO}(n - p) \simeq \iota(\mathsf{SO}(n - p))$ is a subgroup of $\mathsf{SO}(n)$, where,

$$\iota : \mathsf{SO}(n - p) \to \mathsf{SO}(n)$$

defined by

$$X \mapsto \begin{bmatrix} I_p & 0 \\ 0 & X \end{bmatrix}$$

is an isomorphism from $\mathsf{SO}(n - p)$ to $\iota(\mathsf{SO}(n - p))$.

**Differential Geometry of the Grassmannian $\mathsf{Gr}(p, n)$:** The Grassmann manifold (or the Grassmannian) is defined as the set of all $p$-dimensional linear subspaces in $\mathbf{R}^n$ and is denoted by $\mathsf{Gr}(p, n)$, where $p \in \mathbf{Z}^+$, $n \in \mathbf{Z}^+$, $n \geq p$. Grassmannian is a symmetric space and can be identified with the quotient space

$$\mathsf{SO}(n)/S\left(O(p) \times O(n - p)\right),$$

where $S\left(O(p) \times O(n - p)\right)$ is the set of all $n \times n$ matrices whose top left $p \times p$ and bottom right $n - p \times n - p$ submatrices are orthogonal and all other entries are 0, and overall the determinant is 1.

A point $\mathcal{X} \in \mathsf{Gr}(p, n)$ can be specified by a basis, $X$. We say that $\mathcal{X} = \text{Col}(X)$ if $X$ is a basis of $\mathcal{X}$, where $\text{Col}(.)$ is the column span operator. It is easy to see that the general linear group $\text{GL}(p)$ acts

isometrically, freely and properly on $\mathsf{St}(p,n)$. Moreover, $\mathsf{Gr}(p,n)$ can be identified with the quotient space $\mathsf{St}(p,n)/\mathsf{GL}(p)$. Hence, the projection map

$$\Pi : \mathsf{St}(p,n) \to \mathsf{Gr}(p,n)$$

is a *Riemannian submersion*, where $\Pi(X) \triangleq \mathrm{Col}(X)$. Moreover, the triplet $(\mathsf{St}(p,n), \Pi, \mathsf{Gr}(p,n))$ is a fiber bundle.

*Horizontal and Vertical Space:* At every point $X \in \mathsf{St}(p,n)$, we can define the *vertical space*, $\mathcal{V}_X \subset T_X\mathsf{St}(p,n)$ to be $\mathrm{Ker}(\Pi_{*X})$. Further, given $g^{\mathsf{St}}$, we define the *horizontal space*, $\mathcal{H}_X$ to be the $g^{\mathsf{St}}$-orthogonal complement of $\mathcal{V}_X$.

*Horizontal lift:* Using the theory of principal bundles, for every vector field $\widetilde{U}$ on $\mathsf{Gr}(p,n)$, we define the *horizontal lift* of $\widetilde{U}$ to be the unique vector field $U$ on $\mathsf{St}(p,n)$ for which $U_X \in \mathcal{H}_X$ and $\Pi_{*X}U_X = \widetilde{U}_{\Pi(X)}$, for all $X \in \mathsf{St}(p,n)$.

*Metric on Gr:* As, $\Pi$ is a Riemannian submersion, the isomorphism $\Pi_{*X}|_{\mathcal{H}_X} : \mathcal{H}_X \to T_{\Pi(X)}\mathsf{Gr}(p,n)$ is an isometry from $(\mathcal{H}_X, g_X^{\mathsf{St}})$ to $(T_{\Pi(X)}\mathsf{Gr}(p,n), g_{\Pi(X)}^{\mathsf{Gr}})$. So, $g_{\Pi(X)}^{\mathsf{Gr}}$ is defined as:

$$g_{\Pi(X)}^{\mathsf{Gr}}(\widetilde{U}_{\Pi(X)}, \widetilde{V}_{\Pi(X)}) = g_X^{\mathsf{St}}(U_X, V_X) \tag{1}$$
$$= \mathrm{trace}((X^TX)^{-1}U_X^TV_X)$$

where, $\widetilde{U}, \widetilde{V} \in T_{\Pi(X)}\mathsf{Gr}(p,n)$ and $\Pi_{*X}U_X = \widetilde{U}_{\Pi(X)}$, $\Pi_{*X}V_X = \widetilde{V}_{\Pi(X)}$, $U_X \in \mathcal{H}_X$ and $V_X \in \mathcal{H}_X$.

We covered the exponential map and the Riemannian metric above, and their explicit formulation for manifolds listed above is provided for easy reference in Table 1.

| | $g_X(U,V)$ | $\mathrm{Exp}_X(U)$ | $\mathrm{Exp}_X^{-1}(Y)$ |
|---|---|---|---|
| $\mathsf{St}(p,n)$ Kaneko et al. (2012) | $\mathrm{trace}\left(U^TV\right)$ | $\widetilde{U}\widetilde{V}^T,$ $\widetilde{U}S\widetilde{V}^T = \mathrm{svd}(X+U)$ | $(Y-X) - X(Y-X)^TX$ |
| $\mathsf{Gr}(p,n)$ Absil et al. (2004) | $\mathrm{trace}\left(\Pi_*^{-1}(U)^T\Pi_*^{-1}(V)\right)$ | $\widehat{U}\widehat{V}^T,$ $\widehat{U}\widehat{S}\widehat{V}^T = \mathrm{svd}(\bar{X}+U)$ | $\bar{Y}\left(\bar{X}^T\bar{Y}\right)^{-1} - \bar{X},$ $X = \Pi(\bar{X}), Y = \Pi(\bar{Y})$ |
| $\mathsf{SO}(n)$ Subbarao & Meer (2009) | $\mathrm{trace}\left(X^TUX^TV\right)$ | $X\,\mathrm{expm}\left(X^TU\right)$ | $X\,\mathrm{logm}\left(X^TY\right)$ |

Table 1: Explicit forms for some operations we need. $\Pi(X)$ returns $X$'s column space; $\Pi_*$ is $\Pi$'s differential.

## 1.2 Proof of Theorem 1

We first restate the assumptions from section 2:

**Assumptions:**

**(a)** The random variables $\mathbf{X} \sim \mathcal{N}(\mathbf{0}, \Sigma_x)$ and $\mathbf{Y} \sim \mathcal{N}(\mathbf{0}, \Sigma_y)$ with $\Sigma_x \preceq cI_d$ and $\Sigma_y \preceq cI_d$ for some $c > 0$.

**(b)** The samples $X$ and $Y$ drawn from $\mathcal{X}$ and $\mathcal{Y}$ respectively have zero mean.

**(c)** For a given $k \leq d$, $\Sigma_x$ and $\Sigma_y$ have non-zero top-$k$ eigen values.

Recall that $F$ and $\widetilde{F}$ are the optimal values of the true and approximated CCA objective in (1) and (4) respectively, we next restate Theorem 1 and give its proof:

**Theorem 1.** *Under the assumptions and notations above, the approximation error $E = \|F - \widetilde{F}\|$ is bounded and goes to zero while the whitening constraints in (4b) are satisfied.*

*Proof.* Let $U^*, V^*$ be the true solution of CCA. Let $U = \widetilde{U}S_uQ_u, V = \widetilde{V}S_vQ_v$ be the solution of (4) with $\widetilde{U}, \widetilde{V}$ be the PCA solutions of $X$ and $Y$ respectively with $S_uQ_u = \widetilde{U}^TU^*$ and $S_vQ_v = \widetilde{V}^TV^*$ (using RQ decomposition). Let $\widehat{X} = X\widetilde{U}\widetilde{U}^T$ and $\widehat{Y} = Y\widetilde{V}\widetilde{V}^T$ be the reconstruction of $X$ and $Y$ using principal vectors.

Then, we can write

$$\widetilde{F} = \text{trace}\left(U^T C_{XY} V\right) = \text{trace}\left(\frac{1}{N}\left(\widehat{X}U^*\right)^T \widehat{Y}V^*\right)$$

Similarly we can write $F = \text{trace}\left(\frac{1}{N}\left(XU^*\right)^T YV^*\right)$.

Using Def. 1, we know $\widehat{X}, \widehat{Y}$ follow sub-Gaussian distributions (such an assumption is common for such analyses for CCA as well as many other generic models).

Consider the approximation error between the objective functions as $E = |F - \widetilde{F}|$. Due to von Neumann's trace inequality and Cauchy–Schwarz inequality, we have

$$E = \frac{1}{N}\left|\text{trace}\left((U^*)^T \widehat{X}^T \widehat{Y}(V^*) - (U^*)^T X^T Y (V^*)\right)\right|$$

$$\leq \left|\text{trace}\left((U^*)^T \left(\left(\widehat{X} - X\right)^T \left(\widehat{Y} - Y\right) - 2X^T Y + X^T \widehat{Y} + \widehat{X}^T Y\right)(V^*)\right)\right|$$

$$\leq \sum_i \sigma_i(\widehat{X}_u - X_u)\sigma_i(\widehat{Y}_v - Y_v) + \sum_i \sigma_i(\widehat{X}_u - X_u)\sigma_i(Y_v) + \sum_i \sigma_i(\widehat{Y}_v - Y_v)\sigma_i(X_u)$$

$$\leq \|\left(\widehat{X}_u - X_u\right)\|_F \|\left(\widehat{Y}_v - Y_v\right)\|_F + \|\left(\widehat{X}_u - X_u\right)\|_F \|Y_v\|_F + \left(\widehat{Y}_v - Y_v\right)\|_F \|X_u\|_F$$

$$\text{(A.1)}$$

Here $A_u = AU^*$ and $A_v = AV^*$ for any suitable $A$. where $\sigma_i(A)$ denote the $i$-th singular value of matrix A and $\| \bullet \|_F$ denotes the Frobenius norm.

Now, using Proposition 1, we get

$$\|\left(\widehat{X}_u - X_u\right)\|_F \leq \min\left(\sqrt{2k}\|\Delta_x\|_2, \frac{2\|\Delta_x\|_2^2}{\lambda_k^x - \lambda_{k+1}^x}\right)$$

$$\|\left(\widehat{Y}_v - Y_v\right)\|_F \leq \min\left(\sqrt{2k}\|\Delta_y\|_2, \frac{2\|\Delta_y\|_2^2}{\lambda_k^y - \lambda_{k+1}^y}\right) \quad \text{(A.2)}$$

where,

$$\Delta_x = C(X_u) - C(\widehat{X}_u) \quad \Delta_y = C(Y_v) - C(\widehat{Y}_v). \quad (2)$$

Here $\lambda^x$s and $\lambda^y$s are the eigen values of $C(X_u)$ and $C(Y_v)$ respectively. Now, assume that $C(X_u) = I_k$ and $C(Y_v) = I_k$ since $X_u$ and $Y_v$ are solutions of Eq. 1. Furthermore assume $\lambda_k^x - \lambda_{k+1}^x \geq \Lambda$ and $\lambda_k^y - \lambda_{k+1}^y \geq \Lambda$ for some $\Lambda > 0$. Then, we can rewrite equation A.1 as

$$E \leq \min\left(\sqrt{2k}\|I_k - C(\widehat{X}_u)\|_2, \frac{2\|I_k - C(\widehat{X}_u)\|_2^2}{\Lambda}\right) \min\left(\sqrt{2k}\|I_k - C(\widehat{Y}_v)\|_2, \frac{2\|I_k - C(\widehat{Y}_v)\|_2^2}{\Lambda}\right) +$$

$$\min\left(\sqrt{2k}\|I_k - C(\widehat{X}_u)\|_2, \frac{2\|I_k - C(\widehat{X}_u)\|_2^2}{\Lambda}\right)\|Y_v\|_F +$$

$$\min\left(\sqrt{2k}\|I_k - C(\widehat{Y}_v)\|_2, \frac{2\|I_k - C(\widehat{Y}_v)\|_2^2}{\Lambda}\right)\|X_u\|_F$$

As $C(\widehat{X}_u) \to I_k$ or $C(\widehat{Y}_v) \to I_k$, $E \to 0$. Observe that the limiting conditions for $C(\widehat{X}_u)$ and $C(\widehat{Y}_v)$ can be satisfied by the "whitening" constraint. In other words, as $C(X_u) = I_k$ and $C(Y_v) = I_k$, $C(\widehat{X}_u)$ and $C(\widehat{Y}_v)$ converge to $C(X_u)$ and $C(Y_v)$, the approximation error goes to zero. $\square$

### 1.3 Proof of Proposition 4

Here, we prove that the CCA objective function is geodesically convex as a function of $U$. An analogous analysis follows in case of $V$. With a given solution of $V$, we define the objective function as (i.e., the CCA objective) $f : \mathcal{M} \to \mathbf{R}$ given by

$$f(U) = \text{trace}(U^T C_{XY} V) \quad (3)$$

Proceeding, we see $\mathrm{grad}f(U) = C_{XY}V$.

Now, we can show $f$ is geodesically convex by showing

$$f(U_1) - f(U_2) \leq g_{U_1}(\mathrm{grad}f(U_1), -\mathrm{Exp}_{U_1}^{-1}(U2)) \tag{4}$$

for $U1, U2 \in \mathcal{M}$ and $g$ is the Riemannian metric (Bécigneul & Ganea (2018) calls this $\rho$). First, consider the LHS of the inequality.

$$f(U_1) - f(U_2) = \mathrm{trace}(U_1^T C_{XY}V) - \mathrm{trace}(U_2^T C_{XY}V) = \mathrm{trace}((U_1 - U_2)^T C_{XY}V) \tag{5}$$

Now let us calculate $-\mathrm{Exp}_{U_1}^{-1}(U_2)$ which will be used for the RHS of the above inequality for geodesic convexity,

$$-\mathrm{Exp}^{-1}U_1(U_2) = (U_1 - U_2) + U_1(U_2 - U_1)^T U_1 \tag{6}$$

Then, for the RHS of the inequality we just plug in the terms, $g_{U_1}(\mathrm{grad}f(U_1), -\mathrm{Exp}_{U_1}^{-1}(U2)$ equal to

$$\mathrm{trace}((U_1 - U_2)^T C_{XY}V) + \mathrm{trace}(V^T C_{XY}^T U_1(U_2 - U_1)^T U_1) \tag{7}$$

Observe that in order to show

$$f(U_1) - f(U_2) \leq g_{U_1}(\mathrm{grad}f(U_1), -\mathrm{Exp}_{U_1}^{-1}(U2)) \tag{8}$$

since the first terms of the LHS and RHS match, we only need to show $\mathrm{trace}(V^T C_{XY}^T U_1(U_2 - U_1)^T U_1) \geq 0$. Let $\widetilde{U}$ be the solution of PCA, then $\mathrm{trace}(V^T C_{XY}^T \widetilde{U}) > 0$, as $\widetilde{U}$ lies in the feasible set of maximizing correlation, i.e., solution of PCA belongs to the CCA feasible set.

Now let us assume that the consecutive iterates lie inside a geodesic ball of radius less than the convexity radius. Let $U_1 = \widetilde{U}$ and $U_2$ be the solution of second iterate. Then $U_1(U_2 - U_1)^T U_1$ lies inside the geodesic ball of a radius less than the convexity radius as it is linear combination of columns of $U_1$. This ensures that $\mathrm{trace}(V^T C_{XY}^T U_1(U_2 - U_1)^T U_1)$ has the same sign as the term $\mathrm{trace}(V^T C_{XY}^T U_1)$ (using intermediate value theorem on the convex ball), since $\mathrm{trace}(V^T C_{XY}^T U_1(U_2 - U_1)^T U_1)$ is zero only if $U_2 = U_1$, i.e., at convergence.

Thus,

$$f(U_1) - f(U_2) \leq g_{U_1}(\mathrm{grad}f(U_1), -\mathrm{Exp}_{U_1}^{-1}(U2)) \tag{9}$$

and hence it is geodesically convex (compare this inequality with equation (12) of Bécigneul & Ganea (2018)).

## 1.4 Implementation details of CCA on fixed dataset

**Implementation details.** On all three benchmark datasets, we only passed the data once for both our RSG+ and MSG Arora et al. (2017) and we use the code from Arora et al. (2017) to produce MSG results. We conducted experiments on different dimensions of target space: $k = 1, 2, 4$. The choice of $k$ is motivated by the fact that the spectrum of the datasets decays quickly. Since our RSG+ processes data in small blocks, we let data come in mini-batches (mini-batch size was set to 100).

## 1.5 Runtime of RSG+ and baseline methods

In addition to the runtime comparison between RSG+ and MSG, we also plot the runtime of our algorithm under different data dimension (set $d_x = d_y = d$) and number of total samples sampled from joint gaussian distribution in Fig. 2.

## 1.6 Error metrics for fairness

**Equality of Opportunity (EO) Hardt et al. (2016)**: A classifier $h$ is said to satisfy EO if the prediction is independent of the protected attribute $s$ (in our experiment $s$ is a binary variable where $s = 1$ stands for *Male* and $s = 0$ stands for *Female*) for classification label $y \in \{0, 1\}$. We use the

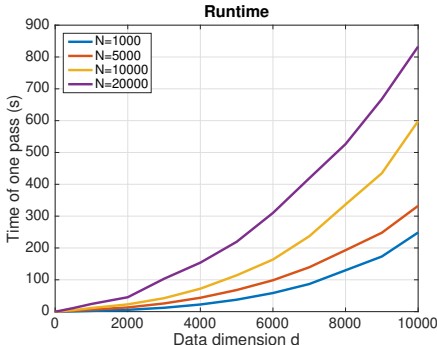

Figure 2: Runtime of RSG+ under different data dimensions and size of datasets.

difference of false negative rate (conditioned on $y = 1$) across two groups identified by protected attribute $s$ as the error metric, and we denote it as DEO.

**Demographic Parity (DP) Yao & Huang (2017)**: A classifier $h$ satisfies DP if the likelihodd of making a misclassification among the positive predictions of the classifier is independent of the protected attribute $s$. We denote the difference of demographic parity between two groups identified by the protected attribute as DDP.

### 1.7 Implementation details of fairness experiments

**Implementation details.** The network is trained for 20 epochs with learning rate 0.01 and batch size 256. We follow Donini et al. (2018) to use NVP (novel validation procedure) to evaluate our result: first we search for hyperparameters that achieves the highest classification score and then report the performance of the model which gets minimum fairness error metrics with accuracy within the highest $90\%$ accuracies. When we apply our RSG+ on certain layers, we first use randomized projection to project the feature into 1k dimension, and then extract top-10 canonical components for training. Similar to our previous experiments on DeepCCA, the batch method does not scale to 1k dimension.

**Resnet-18 architecture and position of Conv-0,1,2 in Table 3.** The Resnet-18 contains a first convolutional layer followed by normalization, nonlinear activation, and max pooling. Then it has four residual blocks, followed by average polling and a fully connected layer. We denote the position after the first convolutional layer as conv0, the position after the first residual block as conv1 and the position after the second residual block as conv2. We choose early layers since late layers close to the final fully connected layer will have feature that is more directly relevant to the classification variable (*attractiveness* in this case).

Table 2: Results of Yger et al. (2012) (on CIFAR-10, our implementation of Yger et al. (2012) faces convergence issues).

| Performance | MNIST | | | Mediamill | | |
|---|---|---|---|---|---|---|
| | $k = 1$ | $k = 2$ | $k = 4$ | $k = 1$ | $k = 2$ | $k = 4$ |
| PCC | 0.93 | 0.81 | 0.53 | 0.55 | 0.61 | 0.51 |
| Time (s) | 575.88 | 536.46 | 540.91 | 41.89 | 28.66 | 28.76 |

### 1.8 Comparison with Yger et al. (2012)

We implemented the method from Yger et al. (2012) and conduct experiments on the three datasets above. The results are shown in Table 2. We tune the step size between $[0.0001, 0.1]$ and $\beta = 0.99$ as used in their paper. On MNIST and MEDIAMILL, the method performs comparably with ours except $k = 4$ case on MNIST where it does not converge well. Since this algorithms also has a $d^3$ complexity, the runtime is $100\times$ more than ours on MNIST and $20\times$ more on Mediamill. On CIFAR10, we fail to find a suitable step size for convergence.