# OpenReview forum: "An Online Riemannian PCA for Stochastic Canonical Correlation Analysis"
_NeurIPS.cc/2021/Conference — NeurIPS 2021 Poster_

### Official Review · Reviewer_P3Jd · 2021-07-13

**Rating:** 7
**Confidence:** 2

**Summary:**

This paper studies the problem of canonical correlation analysis (CCA) to find linear combinations that maximize the correlation between two random variables. Following a reformulation of the problem based on on the principal component analysis (PCA) solution, the authors propose a stochastic algorithm for CCA, called RSG+. Then, they show that their approach improves existing algorithms by achieving lower runtime complexity and/or better convergence rate. Finally, they compare the performance of their algorithm on different image datasets.

**Limitations And Societal Impact:**

The authors briefly discuss in the conclusions a potentially negative societal impacts if CCA is used for discriminative purposes. Similarly to the authors, I do not see a specific negative impact of the authors' approach.

**Main Review:**

## Originality

The main claim of the authors is to achieve significantly lower time complexity per iteration for solving the CCA problem while maintaining $\mathcal{O}(1/t)$ convergence.
The authors reformulate CCA using the solution to principal component analysis. Later, they show with Theorem 1 that the error between the reformulated and the original problem is bounded and goes to zero.
All the convergence properties (Prop. 2-4) of the algorithm presented by the authors are proved using properties available in the literature, which the authors reference.
Could the authors clarify the novelty of their approach, given that the convergence of the algorithm has already been proved in the literature ?

## Quality

The proof of Theorem 1 uses Def. 1 and Prop. 1 which are valid for subgaussian variables, while the assumption of Theorem 1 requires Gaussian variables for X and Y.
Is it possible to generalize Theorem 1 and the later statements in section 2.1 to subgaussian variables ?
Also, the introduction and section 2 introduce the problem for general random variables while the analysis in the paper seems to hold only for gaussian (or sub-gaussian variables).

Based on l.219-225, it seems that the most expensive part of the algorithm is to compute the SVD of matrices, which can be very expensive for large matrices. Would it be possible to further reduce the computational expense of the algorithm using the randomized SVD algorithm ?

Section 3.1: I cannot find the table reporting the runtime improvement over MSG, can the authors reference the table precisely.
Table 2 in the Supplementary Material is not clear (the caption is not detailed) and only references one runtime, is it for the authors' algorithm, MSG ?
Is there an explanation for why RSG+ underperforms when the top k eigenvalues are dominated by the top l eigenvalues ?

Section 3.2: The results from the experiment are not clear, do the authors claim that they maintain the same accuracy as DeepCCA while being able to scale to larger d ?

## Clarity

The paper is well written and the authors start by introducing the CCA problem and stating current challenges and their contributions clearly, which is very appreciable.
The structure of the paper is clear and the authors first go through the theoretical details, then describe their algorithm, and consider different numerical examples to evaluate it in practice.

l.76 I would appreciate more explanations/intuitions to motivate this reformulation
l.110 can the authors define the notation $\Sigma_x \leq cI_d$, does it mean that $c I_d-\Sigma_x$ is positive definite ? Why is this assumption needed ?
The statement of Thm 1 is not clear, can the authors specify the quantity that converges so that the error goes to 0 ?

Would it not make sense to write down the bound explicitly in the statement of the theorem 1 (at least in the supplementary material) ?

The paper would gain in clarity by adding a paragraph in the introduction about convergence rate (introducing the notion and reviewing the literature).

Minor comments:
- In the proof of Theorem 1, should the solution be denoted as $U$,$V$ instead of $U^\*$,$V^\*$ (following eq. (1)) ?
- l. 63 of the Supplementary Material, typo for "A. where"
- Throughout the paper, "eigen values" should read "eigenvalues"

## Significance

The contributions of this paper is significant as the algorithm proposed by the author has a lower time-complexity and a better convergence rate than alternating approaches. Therefore, it allows to scale in large dimensions.

**Time Spent Reviewing:**

4 hours

---

> ### Author Response · Authors · 2021-08-10
> **Author Response for Reviewer P3Jd**
>
> We thank the reviewer for reading the paper and asking for clarifications. We address all doubts below. Thanks.
>
> Q1: Could the authors clarify the novelty of their approach, given that the convergence of the algorithm has already been proved in the literature ?
>
> Response: The key novelty of our paper is to demonstrate that the reformulation presented here allows identifying a set of core computational tasks that can benefit from battle tested routines developed for numerical optimization on manifolds. This perspective allows porting over available results to the stochastic variant of the CCA problem. Yes, we agree that this paper does not derive convergence results of the sub-modules but believe there is significant value in knowing that these results do carry over. In the end, the key deliverable/takeaway is an efficient algorithm for stochastic CCA with desirable properties and a favorable performance profile which much broadens the scope for CCA’s applicability beyond what is offered by deepCCA.
>
> Q2: Is it possible to generalize Theorem 1 and the later statements in section 2.1 to sub-gaussian variables ? Also, the introduction and section 2 introduce the problem for general random variables while the analysis in the paper seems to hold only for gaussian (or sub-gaussian variables).
>
> Response: We want to point out that the CCA algorithm presented in Algorithm 1 does not require any assumption on the data distribution. But the analysis of convergence requires the sub-Gaussian assumption. This assumption is mainly to bound the PCA error and is a common assumption when extracting PCs from samples. Thus, for convergence analysis, the sub-Gaussian assumption is important and we do not know of a simple way to avoid it.
>
> Theorem 1 and statements in section 2.1 in fact hold for the sub-Gaussian assumption as well. We will clarify this in the revision.
>
> Q3: Based on l.219-225, it seems that the most expensive part of the algorithm is to compute the SVD of matrices, which can be very expensive for large matrices. Would it be possible to further reduce the computational expense of the algorithm using the randomized SVD algorithm ?
>
> Response: Yes, we agree that a randomized SVD algorithm is quite useful in practice, and our public codebase will include this feature. This was not included as an intrinsic component of the main algorithm to avoid additional bookkeeping in the analysis. Furthermore, the randomization although can be helpful to alleviate the complexity for large matrices, can potentially affect the convergence rate and hence is omitted for the sake of analysis.
>
> Q4: Section 3.1: I cannot find the table reporting the runtime improvement over MSG, can the authors reference the table precisely. Table 2 in the Supplementary Material is not clear (the caption is not detailed) and only references one runtime, is it for the authors' algorithm, MSG ? Is there an explanation for why RSG+ underperforms when the top k eigenvalues are dominated by the top l eigenvalues ?
>
> Response: Thanks a lot for pointing this out. Our table which includes the runtime of RSG+ and MSG was accidentally left out of the supplement in the last latex compilation and went unnoticed. The table is here and we will add it to the supplement:
>
> |            |       MNIST       | MEDIAMILL        | CIFAR                |
> |------------|-------------------|-------------------|----------------------|
> | Time(s)    | k=1 ; k=2 ; k=4   | k=1 ; k=2 ; k=4   | k=1   ; k=2  ; k=4   |
> | RSG+(Ours) | 4.16; 4.24; 4.71  | 1.89; 1.60; 1.44  | 14.80 ;17.22 ;22.10  |
> | MSG        | 35.32;42.90;49.17 | 11.59;14.21;17.34 | 80.21 ;100.80;106.55 |
>
> Table 2 in the supplement is the runtime of [10] (not MSG), which is an earlier work on computing CCA with $d^3$ complexity. Thus we also compare the runtime with [10].
>
> Q5: Section 3.2: The results from the experiment are not clear, do the authors claim that they maintain the same accuracy as DeepCCA while being able to scale to larger d ?
>
> Response: Yes this is a key message: DeepCCA utilizes the closed form solution and thus incurs a O(d^3) memory cost, while our method only costs O(d^2).
>
> Q6: l.76 I would appreciate more explanations/intuitions to motivate this reformulation l.110 can the authors define the notation $\sum_x \leq cI_d$ , does it mean that $cI_d - \sum_x$  is positive definite ? Why is this assumption needed ? The statement of Thm 1 is not clear, can the authors specify the quantity that converges so that the error goes to 0 ?
>
> Response: Yes, thank you for the opportunity to provide this clarification. $\Sigma_x \leq cI_d$ implies $cI_d - \Sigma_x$ is positive-definite. This essentially implies that the covariance matrix is upper bounded, i.e., the diagonally dominant covariance matrix has diagonal entries that are bounded. The bounded covariance is needed to bound the PCA error and in turn CCA approximation error.
>
> We apologize. In Theorem 1, we want to show the convergence of the CCA approximated error. Observe that CCA error is a function of the number of samples, $N$. Theorem 1 states that the error sequence (CCA approximation error as a function of number of samples) goes to zero asymptotically.
>
> Q7: Would it not make sense to write down the bound explicitly in the statement of the theorem 1 (at least in the supplementary material) ?
>
> Response: Yes, we give the upper bound on the error $E$ in equation following line 69 (in the supplement).
>
> Q8: The paper would gain in clarity by adding a paragraph in the introduction about convergence rate (introducing the notion and reviewing the literature).
>
> Response: Yes, we completely agree. We will be happy to add a brief discussion regarding convergence rate in the revision.

---

> > ### Comment · Reviewer_P3Jd · 2021-08-23
> > **Response to the authors**
> >
> > Thanks to the authors for the detailed response which addresses most of my comments.
> > I'm keeping my score and am happy to recommend acceptance provided the authors do the required clarifications (for Theorem 1 and about convergence rate) in the revision.

---

> > > ### Author Response · Authors · 2021-08-25
> > > **Thanks to the Reviewer P3Jd**
> > >
> > > Thanks to the Reviewer for reading our response and we will do the required clarifications in our revision. If there is anything else we can answer, please let us know.

---

### Official Review · Reviewer_LueT · 2021-07-14

**Rating:** 6
**Confidence:** 4

**Summary:**

This paper studies a Riemannian framework for stochastic CCA.  In particular, they reformulate the problem into an optimization problem over the Stiefel manifold, special orthogonal group, and triangular scaling matrices. The estimates on this product manifold are updated using Riemannian gradient descent. Assuming the iterates are restricted to a strongly convex geodesic ball, then the method converges with rate $O(1/t)$. In practice, the method performs well on a few real data experiments including baseline datasets, deep CCA, as well as an application in fairness.

**Limitations And Societal Impact:**

Societal impact is well-discussed, but the limitations need more expounding, and in particular, the assumptions needed for convergence in Proposition 4, as well as the non-convexity of the method.

**Main Review:**

Upsides:

+ The reformulation of the CCA problem is novel and interesting. It allows for more scalable and efficiently computable CCA.
+ Citations appear adequate.
+ The general idea of the paper is clear.
+ The method performs quite well in the given experiments. In particular, the scaling of deep CCA is interesting, as well as the application to fairness.
+ This is a novel manifold optimization application to CCA. In particular, it takes advantage of the fact that remaining on the manifold keeps the constraints satisfied. Furthermore, it casts the problem as an optimization over a product manifold, which is interesting.

Downsides:

- The manifold optimization techniques and results are just standard applications of known theorems.
-The cited theorems seem to need the objective to be geodesically convex. Is it obvious that this is true for the new CCA objective used here? Furthermore, why are the other assumptions satisfied, i.e., why is the gradient bounded, and why are the iterates bounded?
- It is not clear why the performance degrades for Mediamill. In particular, the authors mention that it is due to the fact that the first eigenvalue is large, but it does not seem to be that much larger.  The data is not ill-conditioned, and it is reasonable that many datasets in the wild may exhibit qualities like this.
- Theorem 1 is not sufficiently clear. The assumptions are not clearly referenced. The use of the norm in Theorem 1 is strange, as it is between 1-dimensional quantities (differences in cost). Also, the statement is not precise, as it is unclear what it means for the quantity to “go to zero” — I don’t see a sequence defined here.
- It is not discussed how step sizes are chosen and how sensitive the method is to choices of these. Also, it seems like initialization may be important, but this is not discussed either.
- How would running projected gradient descent perform on (4)? How much does one gain from Riemannian optimization in practice?

---
After the response, I will raise my score to a 6 since they demonstrate that the function is geodesically convex. However, I think that the argument they use for the boundedness of the iterates needs to also be made more rigorous and included in a revision.

**Time Spent Reviewing:**

2

---

> ### Author Response · Authors · 2021-08-10
> **Author Response for Reviewer LueT**
>
> We thank the reviewer for reading the paper and asking for clarifications. We address all doubts below. Thanks.
>
> Q1: The manifold optimization techniques and results are just standard applications of known theorems.
>
> Response: Yes, we agree. We believe that one strength of this paper is to demonstrate that simply adapting the appropriate (mature) tools from differential geometry can yield an efficient scheme to solve the stochastic form of CCA (and the associated properties we expect of most numerical schemes for statistical model estimation become available with just a bit more work). An interesting feature of the paper pertains to the specific design choices (after numerous unsuccessful attempts) that eventually allows the use of standard results and extremely stable numerical operations to close the loop and obtain an efficiently deployable algorithm.
>
> Q2: The cited theorems seem to need the objective to be geodesically convex. Is it obvious that this is true for the new CCA objective used here? Furthermore, why are the other assumptions satisfied, i.e., why is the gradient bounded, and why are the iterates bounded?
>
> Response:
> Geodesically convex: We are not sure  if the reviewer is referring to a specific theorem in the references or the paper. We do not require the *objective* to be geodesically convex but only need the {$A_t$} (manifold valued variables) in Prop. 4 to lie inside a geodesic ball of radius less than the convexity radius. This assumption is standard for manifold based statistical models due to the presence of varying sectional curvature [1,7,9]. Please let us know if this clarifies the doubt.
>
> Bounded gradient: The objective functions, $\widetilde{F}$ and $\widetilde{F}_{pca}$ are Lipschitz continuous as the population covariance matrices are assumed to be bounded. This implies that the Euclidean gradient is bounded, and as the Riemannian covariant derivative is upper bounded by the Euclidean gradient, we can conclude that a constant to bound gradient in Proposition 4 exists. In practice, this step does not lead to numerical issues in our implementation.
>
> Bounded iterates: This is derived directly by "inverting" the specified tolerance level. Our objective is defined using a finite sum. Since we use a stochastic algorithm, the number of iterations sufficient to reach the tolerance level is based on using the error on iteration $t$ (used in showing the convergence rate) and solving for $t$. In order to get $\epsilon$ tolerance, we need a number of iterates bounded by $O(N + D/\epsilon^2)$ for some constant $D > 0$ which depends on Lipschitz constant for the objective function and the sectional curvature bound of the underlying manifold (denoted by $\kappa$ in Proposition 4). We are happy to provide the detailed expression right after the convergence rate result.
>
> Q3: It is not clear why the performance degrades for Mediamill. In particular, the authors mention that it is due to the fact that the first eigenvalue is large, but it does not seem to be that much larger. The data is not ill-conditioned, and it is reasonable that many datasets in the wild may exhibit qualities like this.
>
> Response: We should clarify that the performance of our method does not actually degrade for Mediamill. In fact, the performance curve of our method in Fig. 2(b) (Mediamill) has a similar trend to the one in Fig. 2(a) (MNIST) and Fig. 2(c) (CIFAR), and achieves a similar final TCC as the one on CIFAR. Compared to MSG [4] on Mediamill, we perform better on k=1, comparable on k=2, and only worse than MSG on k=4. By saying that ``our method underperforms [4] when top-k eigenvalues are dominated by top-l eigenvalues’’, we specifically refer to the k=4 case here. Importantly, since it is clear that most of the correlation is captured by the top-2 eigenvalues, it is not practically useful to do CCA for k=4 here. This is not a deficiency of the algorithm. We hope this addresses the doubt.
>
> Q4: Theorem 1 is not sufficiently clear. The assumptions are not clearly referenced. The use of the norm in Theorem 1 is strange, as it is between 1-dimensional quantities (differences in cost). Also, the statement is not precise, as it is unclear what it means for the quantity to “go to zero” — I don’t see a sequence defined here.
>
> Response: In Theorem 1, the norm only implies the absolute difference between $F$ and $\widetilde{F}$. Observe that as the number of samples goes to $\infty$, we showed that asymptotically $E = |F - \widetilde{F}|$ goes to zero. Here, the sequence is the CCA error after inspecting $N$ samples. We will make this point clearer in the revision.
>
> Q5: It is not discussed how step sizes are chosen and how sensitive the method is to choices of these. Also, it seems like initialization may be important, but this is not discussed either.
>
> Response: The step sizes are chosen from {1, 0.1, 0.01, 0.001, 0.0001, 0.00001} using standard cross validation. When using CCA for deep learning (especially the section on fairness), we tune the step size together with the learning rate of the neural network. The initialization of our method only involves the PCA module (we do not have any other initialization steps). We provide some of these details in section 2 but will be happy to add any additional information as suggested.
>
> Q6: How would running projected gradient descent perform on (4)? How much does one gain from Riemannian optimization in practice?
>
> Response: Most, if not all, projected gradient descent (PGD) approaches use *extrinsic* operators compared to *intrinsic* manifold operators as is the case in Riemannian gradient descent. This results in a comparatively slower convergence rate in the case of PGD [see Bhattacharya and Patrangenaru, Annals of Statistics, Large sample theory of intrinsic and extrinsic sample means on manifolds: I and II, 2005]. We also performed some quantitative evaluation on the MNIST dataset. We ran projected gradient descent on (4) (also using PCA as an initialization), with the same setup as discussed in Section 3.1. We run PGD until we achieve convergence. The comparative performance is: when k=1, TCC= 0.482 (ours is 0.831); when k=2, TCC=0.375 (ours is 0.812); when k=4, TCC=0.519 (Ours is 0.762). This clearly demonstrates that in this context, empirically, PGD has a slower convergence rate.

---

> > ### Comment · Reviewer_LueT · 2021-08-31
> > **Some further questions**
> >
> > I am still trying to understand the theoretical contribution of the paper and would like some clarification on the following points.
> >
> > 1) Proposition 4 cites [27] and [5]. Maybe I am missing something, but [5] assumes the functions are geodesically convex over $M$. [27] assumes that the expected function is convex over the set (this does not discuss the manifold case). How then is the result in Proposition 4 straightforward? Where is its proof?
> > 2) The answer in terms of bounded iterates does not answer my question. I was asking why one can assume that the iterates of the algorithm lie within a geodesically convex ball.

---

> > > ### Author Response · Authors · 2021-09-01
> > > **Response to LueT**
> > >
> > >  Please see the response below.
> > >
> > > 1. Proposition 4 cites [27] and [5]. Maybe I am missing something, but [5] assumes the functions are geodesically convex over . [27] assumes that the expected function is convex over the set (this does not discuss the manifold case). How then is the result in Proposition 4 straightforward? Where is its proof?
> > >
> > > Ans: Here, we will prove that the CCA objective function is geodesically convex as a function of $U$. An analogous analysis follows in case of $V$.  With a given solution of $V$, we define the objective function as (i.e., the CCA objective)
> > > $
> > > f:\mathcal{M}\rightarrow \mathbf{R}
> > > $ given by
> > > $$
> > > f(U) = \text{trace}(U^TC_{XY}V).
> > > $$
> > >
> > > Proceeding, we see  $
> > > \text{grad} f(U) = C_{XY}V
> > > $.
> > >
> > > Now, we can show $f$ is geodesically convex by showing
> > >
> > > $$
> > > f(U_1) - f(U_2) \leq g_{U_1}\left(\text{grad}f(U_1), -\text{Exp}^{-1}_{U1}(U_2)\right)
> > > $$
> > >
> > > for $U_1, U_2 \in \mathcal{M}$ and $g$ is the Riemannian metric (as used in our supplement. REF [5] calls this $\rho$). First, consider the LHS of the inequality.
> > >
> > > $$
> > > f(U_1) - f(U_2) = \text{trace}(U_1^TC_{XY}V) - \text{trace}(U_2^TC_{XY}V) = \text{trace}((U_1 - U_2)^TC_{XY}V)
> > > $$
> > >
> > >
> > > Now let us calculate $-\text{Exp}^{-1}{U_1}(U_2) $ which will be used for the RHS of the above inequality for geodesic convexity,
> > >
> > > $$
> > > -\text{Exp}^{-1}_{U1}(U_2) = (U_1 - U_2) + U_1(U_2 - U_1)^TU_1
> > > $$
> > >
> > > Then, for the RHS of the inequality we just plug in the terms, $g_{U1}(\text{grad} f(U_1),  -\text{Exp}^{-1}_{U1}(U_2))$ equal to
> > >
> > > $$
> > > \text{trace}((U_1 - U_2)^TC_{XY}V) + \text{trace}\left(V^TC_{XY}^T U_1(U_2 - U_1)^TU_1\right)
> > > $$.
> > >
> > >
> > > Observe that in order to show
> > >
> > > $$
> > > f(U_1) - f(U_2) \leq g_{U_1}\left(\text{grad}f(U_1), -\text{Exp}^{-1}_{U1}(U_2)\right)
> > > $$
> > >
> > > since the first terms of the LHS and RHS match, we only need to show $\text{trace}\left(V^TC_{XY}^T U_1(U_2 - U_1)^TU_1\right) \geq 0$. Let $\widetilde{U}$ be the solution of PCA, then $\text{trace}(V^TC_{XY}^T \widetilde{U}) > 0$, as $\widetilde{U}$ lies in the feasible set of maximizing correlation, i.e., solution of PCA belongs to the CCA feasible set.
> > >
> > > Now let us assume that the consecutive iterates lie inside a geodesic ball of radius less than the convexity radius. Let  $U_1 = \widetilde{U}$ and $U_2$ be the solution of second iterate. Then, $U_1(U_2 - U_1)^TU_1$ lies inside the geodesic ball of a radius less than the convexity radius as it is linear combination of columns of $U_1$. This ensures that $\text{trace}\left(V^TC_{XY}^T U_1(U_2 - U_1)^TU_1\right)$ has the same sign as the term $\text{trace}(V^TC_{XY}^T U_1)$  (using intermediate value theorem on the convex ball), since
> > >
> > > $\text{trace}\left(V^TC_{XY}^T U_1(U_2 - U_1)^TU_1\right)$ is zero only if $U_2 = U_1$, i.e., at convergence.
> > >
> > > Thus,
> > >
> > > $$
> > > f(U_1) - f(U_2) \leq g_{U_1}(\text{grad} f(U_1),  -\text{Exp}^{-1}_{U1}(U_2))
> > > $$
> > >
> > > and hence it is geodesically convex [compare this inequality with equation (12) of [5]].
> > >
> > > 2. The answer in terms of bounded iterates does not answer my question. I was asking why one can assume that the iterates of the algorithm lie within a geodesically convex ball.
> > >
> > > Ans: In our formulation the iterates will lie inside the geodesic ball of radius less than the minimum of injectivity and convexity radius.
> > > Observe that all our iterates are of the form
> > > $
> > > U = \widetilde{U}S_uQ_u
> > > $
> > > and analogous for $V$ where we take $\widetilde{U}$ to be the solution of the PCA of $X$. The paper discusses how our CCA estimator of the above form, $U = \widetilde{U}S_uQ_u$ and $V = \widetilde{V}S_vQ_v$ is consistent.  Now observe that the iterates of the above form implies that $U$ and $\widetilde{U}$ column spans the same subspace on the Grassmannian, thus the distance between $U$ and $\widetilde{U}$ is less than $\pi/2$.
> > >
> > > Now, we know that $U$ and $\widetilde{U}$ lie on the Stiefel manifold for which the minimum of the injectivity and convexity radius is $\pi/2$ (see Absil, Optimization algorithms on matrix manifolds, pp. 148). This implies that all iterates for our CCA estimator lie inside a geodesic ball.
> > >
> > > In practice, during optimization, in Riemannian gradient descent, by using small step size and bounded gradient (gradient whose length is less than $\pi/2$), we ensure the iterates indeed lie inside the geodesic ball.

---

### Official Review · Reviewer_VGVu · 2021-07-16

**Rating:** 6
**Confidence:** 3

**Summary:**

This paper proposes a stochastic algorithm for canonical correlation analysis (CCA). It is based on a reformulation of CCA so that it can use existing tools in online PCA and manifold optimization. Some convergence analyses are shown, and an abundance of experiments (mostly involving deep neural nets) are used to support the claims.

**Limitations And Societal Impact:**

The authors do mention aspects of societal impact of their work.

**Main Review:**

This paper has the appeal of modernizing CCA computations for large-scale machine learning tasks. However, I find this paper not clearly written: notations are not clearly defined and results are not clearly stated. I have an impression that this paper is rushed. Nevertheless, the experiments and applications seem interesting, so I can imagine that this paper would be much improved from revision and re-organization.

Some major questions:

- The reformulation does include the original constraints $U^T C_X U = I_k$ and $V^T C_Y V = I_k$, so I don't know why exactly the reformulation would help. Is it true that we only need to update the other decision variables and then the constraints for $U$ and $V$ are satisfied automatically? The algorithm seems to suggest so but I don't know why.

- In the analysis, many statistical assumptions are stated, but it is unclear how they are used. Proposition 1 seems to be a deterministic result, so I don't know why sub-Gaussian assumption is made. Theorem 1 is stated in an extremely ambiguous way.  What does it mean when it says "goes to zero"? (in terms of dimension?) Does it hold with high probability?

- Is $Exp$ clearly defined? What is the Riemannian gradient update? This is a major algorithmic component that should be clearly defined and stated.

- The theoretical analysis does not seem to contain much novelty (many are cited), but the abstract suggest otherwise.

**Time Spent Reviewing:**

4

---

> ### Author Response · Authors · 2021-08-10
> **Author Response for Reviewer VGVu**
>
> We thank the reviewer for reading the paper and asking for clarifications. We address all doubts below. Thanks.
>
> Q1: (a) The reformulation does include the original constraints $U_TC_XU = I_k$ and $V_TC_YV = I_k$, so I don't know why exactly the reformulation would help.
>
> Response: Yes, the reviewer is right that the reformulation also includes the original constraints. But due to the reformulation, it becomes much easier to satisfy the constraint because of the *implicit* manifold optimization (we briefly described this in a paragraph starting from line 100 and in section 2.1). We request the reviewer to check if the sub-section together with our explanation in (1b) below helps. Please let us know if this doubt is not fully resolved.
>
> Q1: (b) True that we only need to update the other decision variables and then the constraints for $U$  and $V$ are satisfied automatically? The algorithm seems to suggest so but I don't know why.
>
> Response: Yes, the reviewer is correct. We briefly describe how the constraints are, up to a rescaling, automatically satisfied.
>
> The decision variables in the reformulation are $\widetilde{U}, \widetilde{V}, S_u, S_v, Q_u, Q_v$. Now, we can see how solving for these decision variables satisfies the whitening constraint.
>
> The solution for $\widetilde{U}$ and $\widetilde{V}$ are the principal vectors of $X$ and $Y$ respectively. Hence, $\widetilde{U}^TC_{X}\widetilde{U}$ and $\widetilde{V}^TC_{Y}\widetilde{V}$ are diagonal matrices. Now, let us expand $U^TC_XU$ as $Q_u^TS_u^T\widetilde{U}^TC_X \widetilde{U}S_uQ_u$. Since $S_u$ is upper triangular, we have that $T_u = X\widetilde{U}S_uQ_u$ is upper triangular (similarly, we can define $T_v$). Now, recall the first term of the objective function (4a), $\text{trace}(U^TC_{XY}V)$ which can be rewritten as $\text{trace}(T_u^TT_v)$. Since we are maximizing this term, it is maximized only if $T_u$ and $T_v$ are diagonal matrices. Thus, $U^TC_XU$ are $V^TC_YV$ are diagonal matrices and after the optimization converges, we can scale $S_u$ and $S_v$ accordingly to satisfy the whitening constraint.
>
> Empirically, we found that in all of our experiments, the CCA solution satisfies the whitening constraint up to 1e-3 in terms of Mean Square Error (MSE).
>
> Q2: In the analysis, many statistical assumptions are stated, but it is unclear how they are used.
>
> (a) Proposition 1 seems to be a deterministic result, so I don't know why the sub-Gaussian assumption is made.
>
> Response:
> Proposition 1 gives a deterministic upper bound on the *expected* approximation error using PCA (restated from Proposition 2.2 in [31]). In order to prove Theorem 1, we use Proposition 1 to bound the expected reconstruction error $\|F - \widetilde{F}\|$. Specifically, we used this result to bound the error in Eq. (A.1) in supplement. Observe that we needed to use the sub-Gaussian property to get the desired upper bound (please check Eq. (A.2) in the supplement). This assumption aligns with the one made in [31]. We use the expected reconstruction error with respect to the sub-Gaussian distribution. We also note, which the reviewer may already be aware, that this distribution assumption is common for PCA because under this assumption, we can describe the data distribution completely using first and second order moments.
>
> We realized that we missed the expectation notation when defining $\epsilon_k$ (the expected reconstruction error) in Proposition 1 which may be the source of the confusion. We apologize for this mistake and have fixed it.
>
> Q2: (b) Theorem 1 is stated in an extremely ambiguous way. What does it mean when it says "goes to zero"? (in terms of dimension?) Does it hold with high probability?
>
> Response:
> We are sorry that the reviewer felt that the statement of Theorem 1 was unclear. No, we are not upper bounding the cumulative likelihood of failure. We will clarify below and will make adjustments in the revision.
>
> In Theorem 1, we want to upper bound the error incurred by the approximated CCA solution. In order to do that, our error bound has components from both the PCA error and components from the CC decision variables. Notice that the CCA error bound depends on how far the approximated  projected covariance matrix  (projected on the approximated CCA solution) is from the true covariance $\|C(\widehat{X}_u) - C(X_u)\|$ and $\|C(\widehat{Y}_v) - C(Y_v)\|$ using the norm (please see line 69 in supplement). Now, in order to show that the approximation error goes to zero, we need to show these norms go to zero. To our knowledge, it is standard to make this claim in the asymptotic sense, i.e., as the number of samples go to infinity, the sample covariance estimator converges to the true population covariance. With this in hand, we can asymptotically show that our approximated CCA error bound goes to zero. This is explained in the supplement in the paragraph starting from line 53. In summary, this reassures us that the steps in our algorithm allow, in an asymptotic sense, a meaningful comparison of the population covariance and the estimated covariance.
>
> Q3: Is  $Exp$ clearly defined? What is the Riemannian gradient update? This is a major algorithmic component that should be clearly defined and stated.
>
> Response: We understand that the reviewer is asking whether or not the $Exp$ map is well-defined for the manifolds specifically used in this work. We formally define this with the closed-form expression for the manifolds we use in this paper in Table 1 in the supplement.
>
> If the reviewer is instead asking for the definition of $Exp$, we briefly restate the definition here.
> If $X\in \mathcal{M}$, the Riemannian Exponential map at $X$,
> denoted by $Exp_X: T_X \mathcal{M} \rightarrow \mathcal{M}$ is defined as $\gamma(1)$
> where $\gamma:[0,1]\rightarrow \mathcal{M}$. We can find $\gamma$ by solving the following differential equation:
> $$\gamma(0) = X, (\forall t_0 \in [0,1])\frac{d\gamma}{dt}\Big|_{t=t_0} = U.$$ This is fully consistent with [3] and the literature.
>
> Of course, for the manifolds used in this paper, we do not need to solve the ODE to obtain $Exp$, since we can use the closed form expressions for $Exp$ in Table 1 (supplement).
>
> Q4: The theoretical analysis does not seem to contain much novelty (many are cited), but the abstract suggests otherwise.
>
> Response: We acknowledge that our paper does not derive a new analysis result to show our main technical results. However, we hope that the reviewer agrees that our key result, namely strict improvements in complexity with convergence/consistency statements, leans on multiple upstream arguments. We were excited by how suitable (known) results from a few different literature streams can be brought together nicely to provide a rigorous yet practical end-to-end scheme for the stochastic version of a classical problem. Several steps were more delicate/involved than others but understandably, may appear direct in hindsight. For example, even if we were told that a PCA initialization idea is the right thing to do (akin to its use in Procrustes Flow), it was not clear to us at the outset whether any guarantees for approximate solutions to stochastic CCA would be possible via *some* sequence of intermediary steps. We wanted to convey this sentiment in the abstract but are happy to adjust based on the reviewer’s suggestion.

---

> ### Comment · Area_Chair_HxMg · 2021-08-28
> **Response to the response**
>
> Dear reviewer (VGVu) ,
> Thank you for the detailed review.
> It seems that we have disagreements among the reviewers.
> What is your answer to the response of the authors?
> Would you like to update your score or answer to the other reviewers?

---

> > ### Comment · Reviewer_VGVu · 2021-08-30
> > **Score unchanged**
> >
> > I appreciate the authors' response to my review. It is apparent that they have made tremendous efforts in this paper. However, from my expertise (which is statistics/ML theory), I find it hard to raise the score due to lack of rigor in the theorem statement. I understand that there are good practical contributions, but nevertheless I would not recommend acceptance.

---

> > > ### Author Response · Authors · 2021-08-30
> > > **Response to VGVu**
> > >
> > > Dear Reviewer VGVu,
> > >
> > > While we could not change your assessment, we thank you for reading and responding to the explanation.
> > >
> > > We would much appreciate an indication of which parts of our response fell short in fully resolving the ambiguity. Is it simply a matter of writing the theorem statement differently or including more details from the author feedback into the statement of the result? At this stage, it is not clear what else we could do, rewrite or change. If the problem itself is not interesting or exciting to you, which is quite possible (and we respect that), we would much appreciate knowing that too. Thanks again.

---

> > > > ### Comment · Area_Chair_HxMg · 2021-09-09
> > > > **Response to the response to the response**
> > > >
> > > > Dear Reviewer VGVu,
> > > > Would you be so kind and answer the authors?
> > > > There is still no agreement on this paper, including technical issues so your opinion matters.
> > > >
> > > > Thanks
> > > > AC

---

> > > > > ### Comment · Reviewer_VGVu · 2021-09-12
> > > > > **Response and some thoughts**
> > > > >
> > > > > I have tried to take a second look at other reviewers' responses as well as the paper. I admit that I am not an expert in this field, so my opinions may not be accurate. But here are my thoughts.
> > > > >
> > > > > I agree with the authors and other reviewers that reformulating the CCA optimization problem is interesting and it can lead to potential practical use. This idea itself is a good fit for NeurIPS. On the down side, the theorem statements seem very sloppy to me; moreover, there are technical issues raised by other reviewers (e.g., Reviewer LueT) who I believe are experts in this field.
> > > > >
> > > > > After looking at the responses, it appears that the technical issues can be fixed in a revision. Although I believe that it is best to do a full clean-up and resubmit later, I will go with the majority opinions among reviewers and accept this paper. Lastly, I appreciate the authors' long and dedicated reply to my review.

---

> > > > > > ### Author Response · Authors · 2021-09-12
> > > > > > **Response to VGVu**
> > > > > >
> > > > > > Dear Reviewer VGVu,
> > > > > >
> > > > > > We are deeply grateful that you spent time reading our response. We are delighted that we were able to resolve the doubts. Thanks for your vote of confidence in our work. It will be great if you could adjust the score appropriately.
> > > > > >
> > > > > > We appreciate the comment about the theorem statement. We are happy to make the theorem statements more descriptive, this is a simple fix. Our choice, at submission time, to keep the statements short (and leave the details to the proof) was simply to avoid overwhelming/confusing a reviewer who only has a passing interest in the details of the analysis. Your interest in the detailed form of the theorem is refreshing.
> > > > > >
> > > > > > Thanks again to the reviewers and the AC for their time.

---

> > > > > > > ### Comment · Reviewer_VGVu · 2021-09-12
> > > > > > > **Score adjusted**
> > > > > > >
> > > > > > > I have slightly increased the score so that we can reach a consensus on this paper (hopefully this helps ACs to make decisions).

---

### Official Review · Reviewer_FnHY · 2021-07-17

**Rating:** 7
**Confidence:** 3

**Summary:**

The authors present a new streaming algorithm for canonical correlation analysis using a re-parametrization of the projection matrices. The algorithm enjoys improved computational complexity as well as convergence rate for extracting the top-k canonical components.


**Limitations And Societal Impact:**

- From Figure 2, the proposed algorithm has subpar performance when the sample size is small.
- The fairness application utilizes a pre-trained network. When the network is biased, the linear CCA may not be sufficient to achieve fairness.


**Main Review:**

The main idea of the paper is clear and the theoretical analysis looks solid.
The key technique used in the approach is a re-parameterization of the projection matrices such that efficient off-the-shelf streaming PCA algorithms can be utilized.

Theoretical analysis shows that the algorithm can extract the top-k components at reduced computational complexity, from O(d^3) to O(d^2 k) per iteration where d is the dimensionality.

The authors performed extensive numerical experiments on several real datasets, where the proposed method generally captures a higher proportion of correlations. The authors also demonstrated the utility of the method in feature learning as well as a fairness application.

**Time Spent Reviewing:**

1

---

> ### Author Response · Authors · 2021-08-10
> **Author Response for Reviewer FnHY**
>
> We are grateful to the reviewer for spending time reading and evaluating our paper. If there are any additional questions we can answer, please let us know and we will respond promptly. Thank you.

---

> > ### Comment · Reviewer_FnHY · 2021-08-29
> > **After rebuttal**
> >
> > I've read the responses of the authors and I am happy to recommend acceptance.

---

### Decision · Program_Chairs · 2021-09-27

**Decision:**

Accept (Poster)

**Comment:**

The paper seems strong for both theoretical and practical point of view.
There were few issues with the paper, but it seems that most of them were solved during the rebuttal.
The argument used for the boundedness of the iterates needs to be made more rigorous and included in the camera-ready version.